Singh *et al. Genome Biology*　(2024) 25:49

## METHOD

# RUBICON: a framework for designing efficient deep learning-based genomic basecallers

Gagandeep Singh[1,3], Mohammed Alser[1], Kristof Denolf[3], Can Firtina[1*], Alireza Khodamoradi[3], Meryem Banu Cavlak[1], Henk Corporaal[2] and Onur Mutlu[1*]

*Correspondence:
firtinac@ethz.ch; omutlu@ethz.ch

[1] Department of Information Technology and Electrical Engineering, ETH Zürich, Zürich, Switzerland
[2] Department of Electrical Engineering, Eindhoven University of Technology, Eindhoven, The Netherlands
[3] Research and Advanced Development, AMD, Longmont, USA

## Abstract

Nanopore sequencing generates noisy electrical signals that need to be converted into a standard string of DNA nucleotide bases using a computational step called basecalling. The performance of basecalling has critical implications for all later steps in genome analysis. Therefore, there is a need to reduce the computation and memory cost of basecalling while maintaining accuracy. We present RUBICON, a framework to develop efficient hardware-optimized basecallers. We demonstrate the effectiveness of RUBICON by developing RUBICALL, the first hardware-optimized mixed-precision basecaller that performs efficient basecalling, outperforming the state-of-the-art basecallers. We believe RUBICON offers a promising path to develop future hardware-optimized basecallers.

**Keywords:**  Genomics sequencing, Basecalling, Hardware acceleration, Machine learning, Deep neural network

## Background

The rapid advancement of genomics and sequencing technologies continuously calls for the adjustment of existing algorithmic techniques or the development of entirely new computational methods across diverse biomedical domains [1–13]. Modern sequencing machines [14, 15] are capable of sequencing complex genomic structures and variants with high accuracy and throughput using long-read sequencing technology [16]. Oxford Nanopore Technologies (ONT) is the most widely used long-read sequencing technology [16–21]. ONT devices generate long genomic reads, each of which has a length ranging from a few hundred to a million base pairs or nucleotides, i.e., A, C, G, and T in the DNA alphabet [22–26].

ONT devices sequence a genome by measuring changes to an electrical signal as a single strand of DNA is passed through a nanoscale hole or *nanopore* [27]. The generated noisy electrical signal or *squiggle* is decoded into a sequence of nucleotides using

a computationally expensive step, called *basecalling* [18, 28–31]. Basecallers need to address two key challenges to accurately basecall a raw sequencing input: first, providing accurate predictions of each and every individual nucleotide, as the sensors measuring the changes in electrical current can only measure the effect of multiple neighboring nucleotides together [28], and second, tolerating low signal-to-noise ratio (SNR) caused by thermal noise and the lack of statistically significant current signals triggered by DNA strand motions [29].

Modern basecallers use deep learning-based models to significantly (by at least 10%) improve the accuracy of predicting a nucleotide base from the squiggle compared to traditional non-deep learning-based basecallers [15–17, 30, 32–36]. The success of deep learning in genome basecalling is attributed to the advances in its architecture to model and identify spatial features in raw input data to predict nucleotides. However, we observe the following six shortcomings with the current basecallers [32, 37–44]. First, current state-of-the-art basecallers are slow and show poor performance on state-of-the-art CPU and GPU-based systems, bottlenecking the entire genomic analyses. For example, state-of-the-art throughput optimized basecaller, Dorado-fast, takes ~2.1 h to basecall a 300-Gbps (Giga basepairs) human genome at 3× coverage on a server-grade GPU (NVIDIA A10G [45] GPU with 24GiB DRAM and 16× CPU with 64 GiB DRAM) [46], while the subsequent step, i.e., read mapping, takes only a small fraction of basecalling time (~0.11 h using minimap2 [47]). We observe that basecalling is the single longest stage in the genome sequencing pipeline, taking up to 43% of execution time while the subsequent steps of overlap finding, assembly, read mapping, and polishing take 18%, 4%, <1%, and 35% of execution time, respectively.

Second, for real-time sequencing, high basecalling throughput is a critical factor [7]. In particular, scenarios such as *field sequencing* [39] and *adaptive sampling* [48] necessitate rapid basecalling due to hardware limitations and the need for real-time decision-making. Field sequencing, often conducted in remote or resource-constrained environments, demands immediate basecalling to obtain actionable genomic information swiftly. Conventional high-compute infrastructure is often unavailable or impractical in these settings, underscoring the importance of an efficient basecalling process. Similarly, adaptive sampling protocols, aiming to optimize sequencing output based on real-time analysis of initial sequencing data, require a fast and accurate basecaller to make prompt decisions regarding read continuation or rejection. Also, enhancing the speed and efficiency of basecalling is critical for re-basecalling existing datasets using advanced, higher-accuracy models. By revisiting earlier data with improved basecalling algorithms, researchers can achieve a more precise representation of the genomic sequence. Current basecallers provide a tradeoff between speed and accuracy, often leading to sub-optimal performance in real-time sequencing scenarios.

Third, since basecalling shares similarities with automatic-speech recognition (ASR) task, many researchers have directly adapted established ASR models, such as Quartznet [49], Citrinet [50], and Conformers [51], for basecalling without customizing the neural network architecture specifically for the basecalling problem. Such an approach might lead to higher basecalling accuracy but at the cost of large and unoptimized neural network architecture. For example, Bonito_CTC, an expert-designed convolutional neural network (CNN)-based version of Bonito from ONT, has ~10

million model parameters. We show in the "Effect of pruning" section that we can eliminate up to 85% of the model parameters to achieve a 6.67× reduction in model size without any loss in basecalling accuracy. Therefore, current basecalling models are costly to run, and the inference latency becomes a major bottleneck.

Fourth, modern basecallers are typically composed of convolution layers with skip connections[1] [52] (allow reusing of activations from previous layers) that creates two major performance issues: (a) skip connections increase the data lifetime: the layers whose activations are reused in future layers must either wait for this reuse to occur before accepting new input or store the activations for later use by utilizing more memory. Thus, leading to high resource and storage requirements; and (b) skip connections often need to perform additional computation to match the channel size at the input of the non-consecutive layer, which increases the number of model parameters, e.g., `Bonito_CTC` requires ~21.7% additional model parameters due to the skip connections.

Fifth, current basecallers use floating-point precision (32 bits) to represent each neural network layer present in a basecaller. This leads to high bandwidth and processing demands [53–55]. Thus, current basecallers with floating-point arithmetic precision have inefficient hardware implementations. We observe in the "Effect of quantization" section that the arithmetic precision requirements of current basecallers can be reduced ~4× by adjusting the precision for each neural network layer based on the target hardware and desired accuracy.

Sixth, basecallers that provide higher throughput have lower basecalling accuracy. For example, we show in the "RUBICALL: overall trend" section and Additional file 1: Section S4 that `Bonito_CRF-fast` provides up to 51.65× higher basecalling performance using 36.96× fewer model parameters at the expense of the 5.37% lower basecalling accuracy compared to most accurate basecaller.

These six problems concurrently make basecalling slow, inefficient, and memory-hungry, bottlenecking all genomic analyses that depend on it. Therefore, there is a need to reduce the computation and memory cost of basecalling while maintaining their performance. However, developing a basecaller that can provide fast runtime performance with high accuracy requires a deep understanding of genome sequencing, machine learning, and hardware design. At present, computational biologists spend significant time and effort to design and implement new basecallers by an extensive trial-and-error process.

*Our goal* is to overcome the above issues by developing a comprehensive framework for specializing and optimizing a deep learning-based basecaller that provides high efficiency and performance.

To this end, we introduce `RUBICON`, the first framework for specializing and optimizing a machine learning-based basecaller. `RUBICON` uses two machine learning techniques to develop hardware-optimized basecallers that are specifically designed for basecalling. First, we propose `QABAS`, a quantization-aware basecalling architecture search framework to specialize basecaller architectures for hardware implementation while considering hardware performance metrics (e.g., latency, throughput). `QABAS`

---

[1] A skip connection allows to skip some of the layers in the neural network and feeds the output of one layer as the input to the next layers.

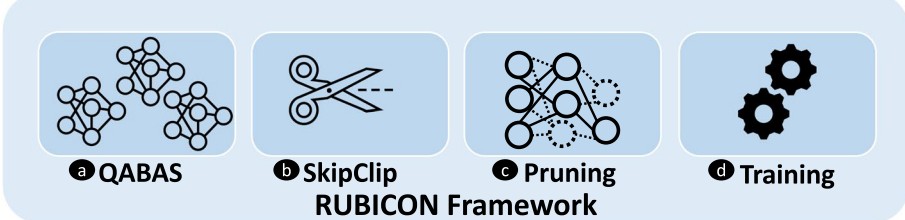

**Fig. 1** Overview of `RUBICON` framework

uses neural architecture search (NAS) [56] to evaluate millions of different basecaller architectures. As discussed in Additional file 1: Section S1, during the basecaller neural architecture search, `QABAS` quantizes the neural network model by exploring and finding the best bit-width precision for each neural network layer, which largely reduces the memory and computational complexity of a basecaller. Adding quantization to the basecaller neural architecture search dramatically increases the model search space ($\sim 6.72\times$ $10^{20}$ more viable options in our search space). However, jointly optimizing basecalling neural network architecture search and quantization allows us to develop accurate basecaller architectures that are optimized for hardware acceleration. Second, we develop `SkipClip` to remove all the skip connections present in modern basecallers to reduce resource and storage requirements without any loss in basecalling accuracy. `SkipClip` performs a skip removal process using knowledge distillation [57], as shown in Additional file 1: Fig. S2 in Additional file 1: Section S2, where we train a smaller network (*student*) without skip connections to mimic a pre-trained larger network (*teacher*) with skip connections. Figure 1 shows the key components of `RUBICON`. It consists of four modules. `QABAS` (ⓐ) and `SkipClip` (ⓑ) are two novel techniques that are specifically designed for specializing and optimizing machine learning-based basecallers. `RUBICON` provides support for `Pruning` (ⓒ), which is a popular model compression technique where we discard network connections that are unimportant to neural network performance [58–61]. We integrate `Training` (ⓓ) module from the official ONT basecalling pipeline [62]. For both the `Pruning` and `Training` modules, we provide the capability to use knowledge distillation [57, 63] for faster convergence and to increase the accuracy of the designed basecalling network.

## Key results

We demonstrate the effectiveness of `RUBICON` by developing `RUBICALL`, the first hardware-optimized mixed-precision basecaller that performs efficient basecalling, outperforming the state-of-the-art basecallers. Additional file 1: Fig. S5 in Additional file 1: Section S2 shows the `RUBICALL` architecture. We compare `RUBICALL` to five different basecallers. We demonstrate six key results. First, `RUBICALL` provides, on average, 2.85% higher basecalling accuracy with 3.77× higher basecalling throughput compared to the fastest basecaller. Compared to an expert-designed basecaller, `RUBICALL` provides 128.13× higher basecalling throughput without any loss in basecalling accuracy by leveraging mixed precision computation when implemented on a cutting-edge spatial vector computing system, i.e., the AMD-Xilinx Versal AIE-ML [64]. Second, we show that `QABAS`-designed models are 5.74× smaller in size with 2.41× fewer neural network

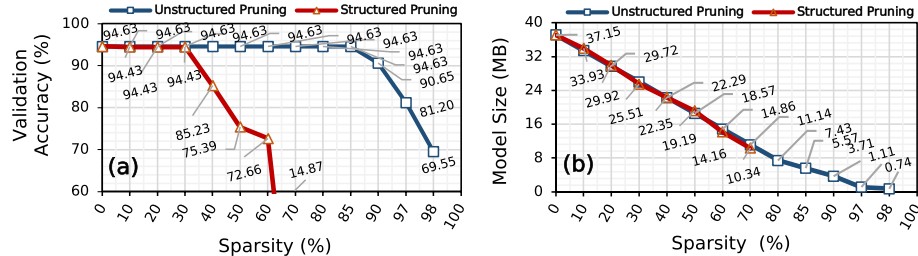

**Fig. 2** Effect of pruning the elements and channels of `Bonito_CTC` using unstructured and structured pruning, respectively, on **a** validation accuracy and **b** model size

model parameters than an expert-designed basecaller. Third, by further using our `Skip-Clip` approach, RUBICALL achieves a 6.88× and 2.94× reduction in neural network model size and the number of parameters, respectively. Fourth, we show in Additional file 1: Section S4 that compared to the most accurate state-of-the-art basecaller (i.e., `Bonito_CRF-sup`), RUBICALL provides 185.54× speedup using 19.22× lower parameters at the expense of, on average, 2.47% lower accuracy. Fifth, assemblies constructed using reads basecalled by RUBICALL lead to higher quality, more contiguous, and more complete assemblies for all evaluated species than that provided by other basecallers. Sixth, RUBICALL provides a 1.82–26.49% lower number of base mismatches with the largest number of mapped bases and mapped reads compared to the baseline basecaller. Our experimental results on state-of-the-art computing systems show that RUBICALL is a fast, memory-efficient, and hardware-friendly basecaller. RUBICON can help researchers develop hardware-optimized basecallers that are superior to expert-designed models and can inspire independent future ideas.

## Results

### Analyzing the state-of-the-art basecaller

We observe established automatic-speech recognition (ASR) models being directly applied to basecalling without optimizing it for basecalling. Such an approach leads to large and unoptimized basecaller architectures. We evaluate the effect of using two popular model compression techniques on the `Bonito_CTC` basecaller: (1) pruning and (2) quantization.

#### *Effect of pruning*

We show the effect of pruning `Bonito_CTC` on the validation accuracy and model size in Fig. 2a and b, respectively. Pruning is a model compression technique where we discard network connections that are unimportant to network performance without affecting the inference accuracy [58–61]. We use unstructured element pruning and structured channel pruning with different degrees of sparsity. Unstructured or element pruning is a fine-grain way of pruning individual weights in a neural network without applying any pruning constraints. While in structured pruning, we remove a larger set of weights while maintaining a dense structure of the model [65, 66].

We make three major observations. First, pruning up to 85% of the `Bonito_CTC` model weights using unstructured pruning reduces the model size by 6.67× while

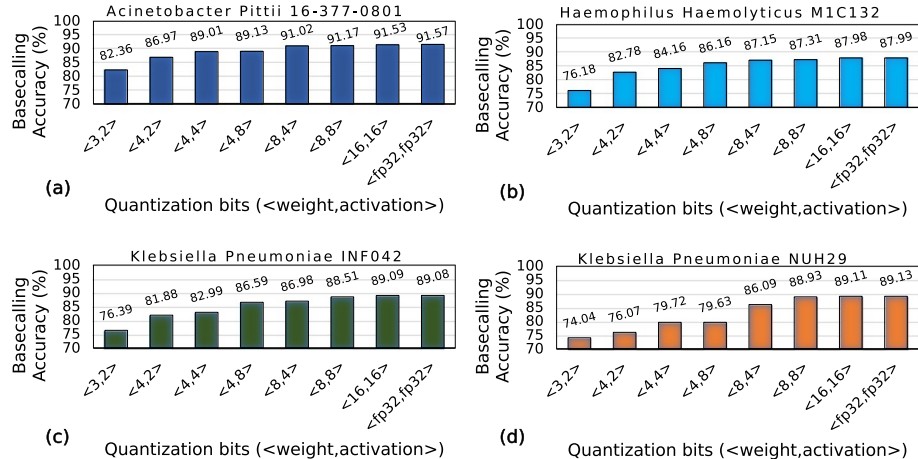

**Fig. 3** Basecalling using quantized models

maintaining the same accuracy as the baseline, unpruned `Bonito_CTC` model. Unstructured pruning leads to the highest model compression [67] at the cost of having sparse weights structure that is unsuitable for acceleration on any hardware platform. While pruning 30–40% of the `Bonito_CTC` model filters, using structured pruning reduces the model size by 1.46–1.66× while maintaining the same accuracy of the baseline, unpruned `Bonito_CTC` model. Such a high pruning ratio shows that most of the weights are redundant and do not contribute to the actual accuracy. Second, after pruning 97% (60%) of the model weights, `Bonito_CTC` provides 81.20% (72.66%) basecalling accuracy while using 33.33× (2.62×) smaller model using unstructured pruning (structured pruning). Third, the *knee point*[2] for unstructured pruning and structured pruning is at 98% and 60% where `Bonito_CTC` provides 65.14% and 72.66% of basecalling accuracy, respectively. Beyond the knee-point, `Bonito_CTC` loses its complete prediction power. We conclude that `Bonito_CTC` is over-parameterized and contains redundant logic and features.

### *Effect of quantization*

Figure 3 shows the effect of using a quantized model to basecall on the basecalling accuracy for four different species. In Fig. 4, we show the effect of quantization on the model size. We quantize both the weight and activation using six different bit-width configurations (`<3,2>`,`<4,2>`,`<4,4>`,`<4,8>`,`<8,4>`, and `<16,16>`). We also show the results with the default floating-point precision (`<fp32,fp32>`). We use static quantization that uses the same precision for each neural network layer.

We make four main observations. First, using a precision of `<8,8>` for weight and activation for all the layers of `Bonito_CTC` causes a negligible accuracy loss (0.18–0.67%), while reducing the model size by 4.03×. Second, `Bonito_CTC` is more sensitive to weight precision than activation precision. For example, we observe a loss of 1.82–9.48% accuracy when using a precision of `<4,8>` instead of `<16,16>` bits compared to an

---

[2] We define knee point as the point beyond which a basecaller is unable to basecall at an acceptable level of accuracy.

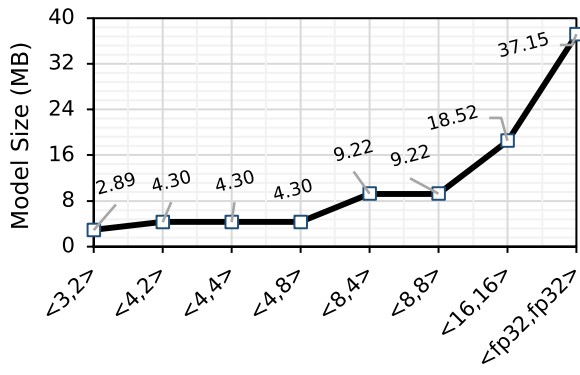

**Fig. 4** Effect of quantizing weight and activation of `Bonito_CTC` on model size. We quantize both the weight and activation with static precision. Since weights are the trainable parameters in a neural network, only weights contribute to the final model size

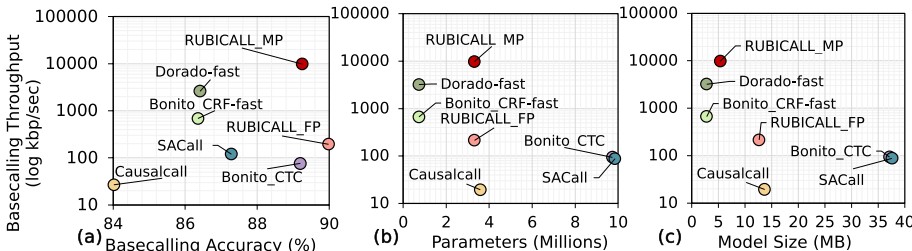

**Fig. 5** Comparison of average basecalling throughput for `RUBICALL-MP` with state-of-the-art basecallers in terms of **a** average basecalling accuracy, **b** model parameters, and **c** model size. `RUBICALL-MP` provides higher compute performance with lower model size when compared to `RUBICALL-FP` because of the mixed-precision computation

accuracy loss of only 0.51–3.02% when using a precision of `<8,4>` instead of `<16,16>` bits. Third, we observe a significant drop in accuracy (by 9.17–15.07%), when using less than 4 bits for weights (e.g., using `<3,2>` configuration). Fourth, using bit-width precision of `<16,16>` bits provides ∼2× reductions in model size and without any accuracy loss compared to using full precision (`<fp32,fp32>`) floating-point implementation. We conclude that the current state-of-the-art basecaller, `Bonito_CTC`, can still efficiently perform basecalling even when using lower precision for both the weight and activation.

### RUBICALL: overall trend

We compare the overall basecalling throughput of `RUBICALL` with that of the baseline basecallers in terms of average basecalling accuracy, model parameters, and model size in Fig. 5a–c, respectively. We evaluate `RUBICALL` using (1) MI210 GPU [68] (`RUBICALL-FP`) using floating-point precision computation and (2) Versal ACAP VC2802 [64], a cutting-edge spatial vector computing system (`RUBICALL-MP`) using mixed-precision computation. The "Methods" section provides details on our evaluation methodology.

We make six key observations. First, compared to `Dorado-fast`, the fastest basecaller, `RUBICALL-MP` provides, on average, 2.85% higher accuracy with 3.77× higher

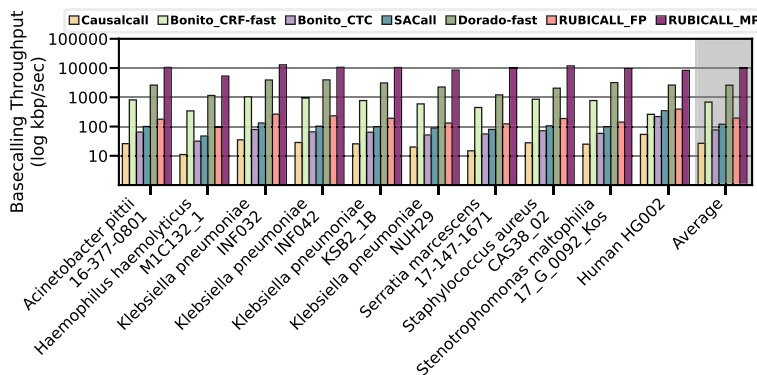

**Fig. 6** Performance comparison of `RUBICALL` (using floating-point precision (`RUBICALL-FP`) and mixed-precision (`RUBICALL-MP`)) and five state-of-the-art basecallers on AMD MI210. The *y*-axis is on a logarithmic scale

basecalling throughput. Therefore, `RUBICALL-MP` provides both accuracy and high basecalling throughput. Second, `RUBICALL-MP` provides 128.13× higher basecalling throughput without any loss in accuracy compared to `Bonito_CTC`, which is an expert-designed basecaller. Unlike `Bonito_CTC`, this is because `RUBICALL-MP` has a mixed precision neural architecture that leads to high compute density. Third, by using mixed-precision quantization, `RUBICALL-MP` provides 50.15× higher performance when compared to its floating-point implementation (`RUBICALL-FP`). Fourth, `SACall` has the highest number of neural network model parameters, which are 2.74×, 13.49×, 1.01×, 13.49×, and 2.97× more than `Causalcall`, `Bonito_CRF-fast`, `Bonito_CTC`, `Dorado-fast`, and `RUBICALL-MP`, respectively. `SACall` uses a large transformer model with an attention mechanism that leads to an over-parameterized model. Fifth, `Dorado-fast` has 4.92×, 13.33×, 13.49×, and 4.54× lower number of trainable model parameters than `Causalcall`, `Bonito_CTC`, `SACall`, and `RUBICALL-MP`. As discussed earlier, `Dorado-fast` provides 2.85% lower accuracy with 3.77× lower basecalling throughput. While `Dorado-fast` has a 4.54× lower number of trainable model parameters, the difference in model size is only 1.92× because `RUBICALL-MP` has each layer quantized to a different precision. Sixth, compared to basecallers with skip connections, `RUBICALL-MP` provides 2.55× and 6.93× smaller model size compared to `Causalcall` and `Bonito_CTC`, respectively. The decrease in model size is due to (1) a lower number of neural network layers and (2) optimum bit-width precision for each neural network layer. Sixth, all the baseline basecallers use floating-point arithmetic precision for all neural network layers. This leads to very high memory bandwidth and processing demands. We conclude that `RUBICALL-MP` provides the ability to basecall quickly and efficiently scale basecalling by providing reductions in both model size and neural network model parameters.

## Performance comparison

We compare the speed of `RUBICALL-MP` against baseline basecallers in Fig. 6. We make three major observations. First, `RUBICALL-MP` consistently outperforms all the other basecallers for all the evaluated species. `RUBICALL-MP` improves average performance by 364.89×, 14.25×, 128.13×, 81.58×, and 3.77× over `Causalcall`, `Bonito_CRF-fast`,

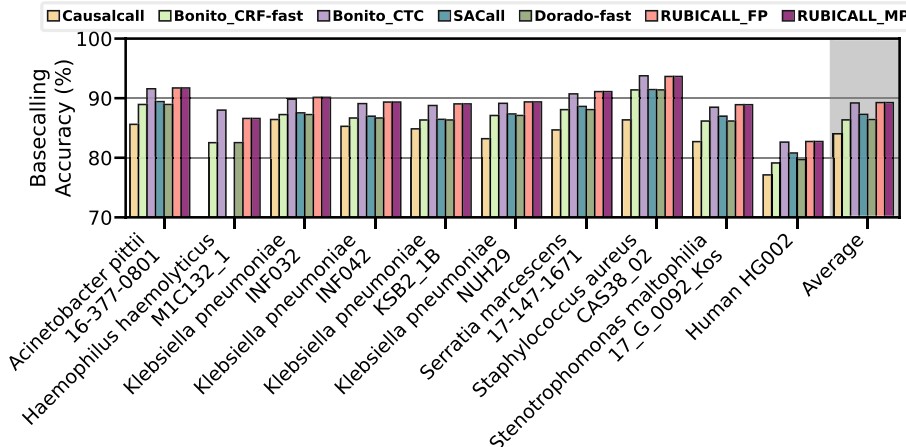

**Fig. 7** Basecalling accuracy comparison of RUBICALL (using floating-point precision (RUBICALL-FP) and mixed-precision (RUBICALL-MP))

Bonito_CTC, SACall, and Dorado-fast, respectively. Second, as RUBICALL-MP each layer is quantized to a different precision, it provides 50.15× higher performance when compared to its floating-point only implementation (RUBICALL-FP). Third, RUBICALL-FP, by using floating-point precision, provides 7.28×, 2.56×, and 1.63× higher performance compared to Causalcall, Bonito_CTC, and SACall, respectively. Additional file 1: Fig. S7 in Additional file 1: Section S5 demonstrates the performance of all the evaluated basecallers on NVIDIA A40 [69] GPU. We conclude that using mixed-precision computation, RUBICALL-MP consistently performs better than the baseline basecallers.

### Basecalling accuracy

We compare the basecalling accuracy of RUBICALL against baseline basecallers in Fig. 7. RUBICALL-MP and RUBICALL-FP use the same model architecture and produce the same basecalled reads, so we report results as RUBICALL. We make three major observations. First, compared to Dorado-fast and Bonito_CRF-fast, we observe RUBICALL achieves 2.85% and 2.89% higher accuracy over these RNN-based basecallers, respectively. RUBICALL provides 5.23% and 0.06% higher accuracy than CNN-based basecaller Causalcall and Bonito_CTC, respectively. Compared to a state-of-the-art transformer-based basecaller, SACall, RUBICALL achieves 1.97% higher basecalling accuracy. Second, Bonito_CTC has 2.93× higher parameters (Fig. 5a) while having similar accuracy as RUBICALL. Third, Causalcall and SACall are unable to align half of *Haemophilus haemolyticus M1C132_1* reads to its reference. Therefore, it is deemed unaligned and cannot be used to determine its read accuracy. We conclude that RUBICALL provides the highest accuracy compared to other basecallers.

### Downstream analysis

#### *De novo assembly*

We provide the statistics related to the accuracy, completeness, and contiguity of assemblies we generate using the basecalled reads from Causalcall, Bonito_CRF-fast, Bonito_CTC, SACall, Dorado-fast, and RUBICALL in Table 1. For Genome

**Table 1** Assembly quality comparison of the evaluated basecallers for different species. We measure assembly accuracy in terms of genome fraction (Genome Fraction (%)) and average identity (Average Identity (%)). Genome fraction is the portion of the `Reference` genome that can align to a given assembly, while average identity is the average of the identity of assemblies when compared to their respective `Reference` genomes. We measure statistics related to the contiguity and completeness of the assemblies in terms of the overall assembly length (Assembly Length), Average GC content (Average GC (%)) (i.e., the ratio of G and C bases in an assembly), NG50 statistics (NG50) (i.e., shortest contig at the half of the overall `Reference` genome length), total number of indels in all aligned bases in the assembly (Total Indels), the ratio of indels to assembly length (Indel Ratio (%)), and the reliability of basepairs using the quality value (Quality Value). NA indicates that the generated assemblies were unalignable to the reference genome

| Dataset | Basecaller | Genome fraction (%) | Average identity (%) | Assembly length | Average GC (%) | NG50 | Total indels | Indel ratio (%) | Quality value (QV) |
|---|---|---|---|---|---|---|---|---|---|
| Acineto-bacter | Causal-call | 92.45 | 86.18 | **3,826,077** | 42.23 | **3,826,077** | 270,228 | 7.06 | 11.99 |
| pittii 16-377-0801 | Bonito_CRF-fast | 96.64 | 89.29 | 3,628,317 | **38.82** | 3,628,317 | 242,373 | 6.68 | 12.03 |
| | Bonito_CTC | **96.87** | 91.44 | 3,676,821 | 38.9 | 3,676,821 | 210,496 | 5.72 | 12.45 |
| | SACall | 96.68 | 89.42 | 3,699,232 | 38.7 | 3,699,232 | 247,997 | 6.7 | 12.1 |
| | Dorado-fast | 96.37 | 88.72 | 3,839,847 | 39.09 | 3,839,847 | 245,016 | 6.38 | 12.03 |
| | RUBICALL | **96.87** | **91.51** | 3,694,086 | **38.82** | 3,694,086 | **208,748** | **5.65** | **15.42** |
| | Reference | 100 | 100 | 3,814,719 | 38.78 | 3,814,719 | 0 | 0 | - |
| Haemophilus | Causal-call | 0.00 | 0.00 | 0 | 0 | 0 | 0 | 0 | NA |
| haemolyticus | Bonito_CRF-fast | 88.76 | **91.51** | **2,046,024** | 37.98 | **2,046,024** | 128,481 | 6.28 | 12.25 |
| M1C132_1 | Bonito_CTC | **96.87** | 90.70 | 1,957,480 | 38.87 | 1,957,480 | **118,253** | **6.04** | 15.34 |
| | SACall | 90.11 | 88.45 | 2,032,994 | **38.22** | 1,880,730 | 134,702 | 6.63 | 13.15 |
| | Dorado-fast | 89.42 | 88.97 | 2,110,860 | 39.49 | 2,110,860 | 129,503 | 6.14 | 12.38 |
| | RUBICALL | **96.87** | 90.54 | 1,966,781 | 38.92 | 1,966,781 | 119,777 | 6.09 | **15.37** |
| | Reference | 100 | 100 | 2,042,591 | 38.46 | 2,042,591 | 0 | 0 | - |
| Klebsiella | Causal-call | 92.45 | 87.35 | **4,959,127** | 56.9 | 4,959,127 | 353,550 | 7.13 | 10.54 |
| pneumoniae | Bonito_CRF-fast | 92.69 | 87.53 | 4,761,297 | **57.19** | 4,761,297 | 347,299 | 7.29 | 10.56 |
| INF032 | Bonito_CTC | 94.50 | 90.20 | 4,897,352 | 56.65 | 4,897,352 | 317,428 | 6.48 | 11.26 |
| | SACall | 93.97 | 88.08 | 4,874,880 | 56.87 | 4,874,880 | 379,028 | 7.78 | 10.8 |
| | Dorado-fast | 93.00 | 87.69 | 5,063,562 | 56.8 | **5,063,562** | 348,572 | 6.88 | 10.64 |
| | RUBICALL | **94.51** | **90.30** | 4,924,240 | 56.85 | 4,924,240 | **314,651** | **6.39** | **11.27** |
| | Reference | 100 | 100 | 5,111,537 | 57.63 | 5,111,537 | 0 | 0 | - |
| Klebsiella | Causal-call | 91.44 | 87.36 | **5,288,166** | **56.94** | **5,288,166** | 374,162 | 7.08 | 10.84 |
| pneumoniae | Bonito_CRF-fast | 92.08 | 88.49 | 5,052,889 | 56.8 | 5,052,889 | 357,354 | 7.07 | 10.93 |
| INF042 | Bonito_CTC | 93.12 | 90.49 | 5,111,083 | 56.61 | 5,111,083 | 317,075 | 6.2 | 11.40 |
| | SACall | 92.93 | 88.60 | 5,149,039 | 56.72 | 5,149,039 | 369,388 | 7.17 | 11.08 |
| | Dorado-fast | 90.21 | 88.20 | 5,737,059 | 56.44 | 5,401,717 | 342,141 | 5.96 | 10.98 |
| | RUBICALL | **93.12** | **90.60** | 5,146,050 | 56.72 | 5,146,050 | **312,448** | **6.07** | **11.42** |
| | Reference | 100 | 100 | 5,337,491 | 57.41 | 5,337,491 | 0 | 0 | - |
| Klebsiella | Causal-call | 91.58 | 86.97 | **5,175,311** | **57.09** | 5,175,311 | 363,807 | 7.03 | 10.88 |
| pneumoniae | Bonito_CRF-fast | 90.24 | 88.00 | 4,932,626 | 56.71 | 4,932,626 | 357,769 | 7.25 | 10.86 |
| KSB2_1B | Bonito_CTC | 93.07 | **90.11** | 5,003,377 | 56.69 | 5,003,377 | **320,519** | **6.41** | **11.41** |

**Table 1** (continued)

| Dataset | Basecaller | Genome fraction (%) | Average identity (%) | Assembly length | Average GC (%) | NG50 | Total indels | Indel ratio (%) | Quality value (QV) |
|---|---|---|---|---|---|---|---|---|---|
| | SACall | **93.58** | 88.19 | 5,034,408 | 56.79 | 5,034,408 | 372,380 | 7.4 | 11.16 |
| | Dorado-fast | 90.28 | 87.67 | 5,442,186 | 56.72 | **5,261,731** | 349,387 | 6.42 | 11.03 |
| | RUBICALL | 93.07 | 89.89 | 5,023,639 | 56.75 | 4,932,626 | 357,769 | 7.12 | 11.25 |
| | Reference | 100 | 100 | 5,228,889 | 57.59 | 5,228,889 | 0 | 0 | - |
| Klebsiella | Causal-call | 89.08 | 86.01 | **5,158,874** | 56.78 | **5,158,874** | 389,676 | 7.55 | 11.75 |
| pneumoniae | Bonito_CRF-fast | 92.17 | 89.34 | 4,942,833 | 57.01 | 4,942,833 | 355,690 | 7.2 | 11.47 |
| NUH29 | Bonito_CTC | **94.36** | 90.26 | 4,918,147 | 57.04 | 4,918,147 | 324,406 | 6.6 | **11.92** |
| | SACall | 93.66 | 88.58 | 4,978,307 | 57.06 | 4,978,307 | 360,950 | 7.25 | 11.56 |
| | Dorado-fast | 92.27 | 88.12 | 5,195,594 | 57.01 | 5,195,594 | 355,728 | 6.85 | 11.56 |
| | RUBICALL | **94.36** | **90.43** | 4,940,813 | 57.18 | 4,940,813 | **316,019** | **6.4** | 11.83 |
| | Reference | 100 | 100 | 5,134,281 | 57.61 | 5,134,281 | 0 | 0 | - |
| Serratia | Causal-call | 89.91 | 86.23 | **5,532,953** | 57.86 | **5,422,052** | 401,545 | 7.26 | **13.39** |
| marcescens | Bonito_CRF-fast | 96.06 | 89.56 | 5,479,812 | **58.85** | 5,282,474 | 345,351 | 6.3 | 12.66 |
| 17-147-1671 | Bonito_CTC | **96.76** | 91.38 | 5,534,329 | 58.41 | 5,316,651 | 298,982 | 5.4 | 13 |
| | SACall | 94.29 | 89.36 | 5,366,913 | 58.57 | 5,366,913 | 358,954 | 6.69 | 12.27 |
| | Dorado-fast | 96.51 | 88.87 | 5,758,989 | 58.29 | 5,282,474 | 348,968 | 6.06 | 12.5 |
| | RUBICALL | **96.76** | **91.59** | 5,597,251 | 58.52 | 5,346,640 | **294,643** | **5.26** | 13.01 |
| | Reference | 100 | 100 | 5,517,578 | 59.13 | 5,517,578 | 0 | 0 | - |
| Staphylococ-cus | Causal-call | 94.35 | 87.29 | 2,849,123 | 36.59 | 2,810,038 | 191,730 | 6.73 | 10.8 |
| aureus | Bonito_CRF-fast | 96.27 | 91.49 | 2,790,895 | 33.05 | 2,752,169 | 149,623 | 5.36 | 11.59 |
| CAS38_02 | Bonito_CTC | **97.03** | **93.57** | 2,858,986 | **32.86** | 2,819,356 | **123,542** | **4.32** | **12.82** |
| | SACall | 95.66 | 91.25 | 2,837,503 | 32.91 | 2,798,079 | 165,200 | 5.82 | 11.57 |
| | Dorado-fast | 96.70 | 91.16 | 2,927,882 | 33.52 | 2,752,169 | 152,216 | 5.2 | 11.64 |
| | RUBICALL | **97.03** | 93.36 | **2,860,885** | 33.24 | **2,821,276** | 124,795 | 4.36 | 12.59 |
| | Reference | 100 | 100 | 2,902,076 | 32.82 | 2,902,076 | 0 | 0 | - |
| Stenotropho-monas | Causal-call | 94.85 | 85.73 | **4,823,177** | 63.66 | **4,823,177** | 366,228 | 7.59 | 11.01 |
| maltophilia | Bonito_CRF-fast | 94.60 | 89.74 | 4,596,898 | **65.5** | 4,596,898 | 337,040 | 7.33 | 11.10 |
| 17_G_0092_Kos | Bonito_CTC | 95.42 | 90.14 | 4,664,226 | 64.82 | 4,664,226 | 298,711 | 6.4 | 11.51 |
| | SACall | 95.28 | 88.50 | 4,672,540 | 64.98 | 4,672,540 | 339,853 | 7.27 | 11.11 |
| | Dorado-fast | 92.99 | 87.70 | 4,854,007 | 63.99 | 4,854,007 | 337,105 | 6.94 | 11.01 |
| | RUBICALL | **95.46** | **90.49** | 4,693,744 | 65.03 | 4,693,744 | **289,073** | **6.16** | **11.63** |
| | Reference | 100 | 100 | 4,802,733 | 66.28 | 4,802,733 | 0 | 0 | - |
| Human | Causal-call | NA | NA | 130,962 | 42.95 | 13,522 | NA | NA | NA |
| HG002 | Bonito_CRF-fast | 0.002 | 92.36 | 119,570,537 | 40.34 | 368,848 | 2860 | **0** | **18.87** |
| | Bonito_CTC | **0.430** | **95.06** | 134,732,516 | **40.86** | 371,590 | 384,243 | 0.29 | 18.58 |
| | SACall | NA | NA | 63,025,520 | 39.87 | 320,873 | NA | NA | NA |
| | Dorado-fast | 0.001 | 93.15 | 121,146,376 | 39.8 | 361,677 | **926** | **0** | 17.46 |
| | RUBICALL | 0.125 | 94.50 | **140,928,248** | 40.99 | **393,950** | 100,256 | 0.1 | 17.81 |
| | Reference | 100 | 100 | 2,947,743,500 | 40.79 | 2,947,743,500 | 0 | 0 | - |

`Fraction (%)`, `Average Identity (%)`, and Quality Value (QV), we highlight the highest achieved value. While for `Assembly Length`, `Average GC (%)`, and `NG50`, we highlight the value closest to the real assembly length. For `Total Indels` and `Indel Ratio (%)`, the best-performing basecaller has the lowest value. We also collect the number of unique k-mers and the frequency of each unique k-mer in a given sequence to perform a comparison of under and over-represented k-mers in Additional file 1: Section S7.

We make six key observations. First, assemblies constructed using reads basecalled by `RUBICALL` provide the best reference genome coverage for *all* datasets ("Genome Fraction" in Table 1). This means that assemblies built using `RUBICALL`-basecalled reads are more complete than assemblies built using reads from other basecallers since a larger portion of the corresponding reference genomes align to their assemblies using `RUBICALL`-basecalled reads compared to that of using reads from other basecallers. Second, assemblies constructed using the `RUBICALL` reads usually have a higher average identity than that of `Causalcall`, `Bonito_CRF-fast`, `Bonito_CTC`, `SACall`, and `Dorado-fast`. These average identity results are tightly in line with the basecalling accuracy results we show in Fig. 7. Although `Bonito_CRF-fast` provides a higher average identity for the *Haemophilus haemolyticus* M1C132_1 dataset (i.e., 91.51%), the genome coverage provided by both `Bonito_CRF-fast` and `Dorado-fast` is 2.2% lower than that provided by `RUBICALL` for the same dataset. This means a large portion of the assembly provided by `Bonito_CRF-fast` has low-quality regions as the reference genome cannot align to these regions due to high dissimilarity. Third, assemblies constructed using the `RUBICALL` reads provide better completeness and contiguity as they have (1) assembly lengths closer to their corresponding reference genomes and (2) higher NG50 results in most cases than those constructed using the `Bonito_CRF-fast` and `Bonito_CTC` reads. Fourth, although `Causalcall` usually provides the best results in terms of the assembly lengths and NG50 results, we suspect that these high NG50 and assembly length results are caused due to highly repetitive and inaccurate regions in these assemblies due to their poor genome fraction and average GC content results. The average GC content of the assemblies constructed using the `Causalcall` reads is significantly distant from the GC content of their corresponding reference genomes in most cases. This poor genome fraction and average GC content results suggest that such large NG50 and assembly length values from `Causalcall` may also be caused by poorly basecalled reads that lead to unresolved repetitive regions (i.e., bubbles in genome assembly graphs) or a strong bias toward certain error types (i.e., homopolymer insertions of a certain base) in the assembly [70, 71]. Fifth, the low total indels and indel ratio (%) for `RUBICALL` in an assembled sequence signify a sequence that closely resembles the expected reference with minimal insertions and deletions (indels). This indicates a well-structured and high-quality assembly. Such assemblies offer a clear and accurate representation of the original sequence, facilitating downstream analyses, gene prediction, functional annotation, and comparative genomics. Sixth, `RUBICALL` consistently provides a higher quality value (QV), indicating a low probability of sequencing errors. Therefore, compared to the other evaluated basecallers, `RUBICALL` has higher reliability of the assembled genome.

We conclude that, in most cases, the reads basecalled by `RUBICALL` lead to higher quality, more contiguous, and more complete assemblies than that provided by other state-of-the-art basecallers, `Causalcall`, `Bonito_CRF-fast`, `Bonito_CTC`, `SACall`, and `Dorado-fast`.

### *Read mapping*

We provide the comparison of `RUBICALL` with `Causalcall`, `Bonito_CRF-fast`, `Bonito_CTC`, `SACall`, and `Dorado-fast` in terms of the total number of base mismatches, the total number of mapped bases, the total number of mapped reads, and the total number of unmapped reads in Fig. 8a–d, respectively. We also show the average read length, the overall number of mapped reads and the mapped bases, and the ratio of the number of mapped bases to the number of mapped reads in Additional file 1: Table S2.

We make five key observations. First, `RUBICALL` provides the lowest number of base mismatches, which are 26.97%, 22.66%, 11.45%, 12.35%, and 23.58% lower compared to `Causalcall`, `Bonito_CRF-fast`, `Bonito_CTC`, `SACall`, and `Dorado-fast`, respectively. This indicates that `RUBICALL` provides more accurate basecalled reads that share large similarity with the reference genome. This is in line with the fact that `RUBICALL` provides the highest basecalling accuracy, as we evaluate in the "Basecalling accuracy" section. Second, `RUBICALL` provides, on average, 22.86%, 0.24%, and 4.77% higher number of mapped bases compared to `Causalcall`, `Bonito_CTC`, and `SACall`, respectively, and only 0.3% and 0.4% lower number of mapped bases when compared to `Bonito_CRF-fast` and `Dorado-fast`, respectively. Mapping more bases to the target reference genome confirms that the careful design and optimizations we perform when building `RUBICALL` have no negative effects on the basecalling accuracy. Third, unlike `Causalcall`, `RUBICALL`, `Bonito_CRF-fast`, `Bonito_CTC`, `SACall`, and `Dorado-fast`, all provide a high number of mapped reads. However, `RUBICALL` is the only basecaller

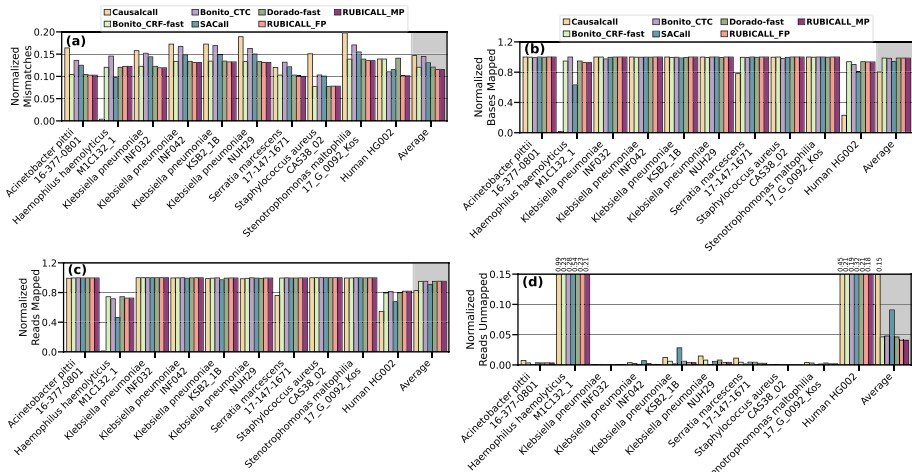

**Fig. 8** Comparison of `RUBICALL` (using floating-point precision (`RUBICALL-FP`) and mixed-precision (`RUBICALL-MP`)) for normalized **a** mismatches, **b** bases mapped, **c** reads mapped, and **d** reads unmapped

that provides high-quality reads that have the highest number of base matches and the lowest number of base mismatches. Fourth, `RUBICALL` achieves 72.66%, 11.79%, 14.63%, 55.02%, and 11.61% lower unmapped reads compared to `Causal-call`, `Bonito_CRF-fast`, `Bonito_CTC`, `SACall`, and `Dorado-fast`, respectively. This indicates that using `Causalcall`, `Bonito_CRF-fast`, `Bonito_CTC`, `SACall`, and `Dorado-fast` wastes a valuable, expensive resource, i.e., sequencing data, by not mapping reads to the reference genome due to basecalling inaccuracies during basecalling. If a read is flagged as unmapped during read mapping, then this read is excluded from all the following analysis steps affecting the overall downstream analysis results. Fifth, for each dataset, we find that the ratio of the number of mapped bases to the number of mapped reads and the average length of the reads are mainly similar across all basecallers (Additional file 1: Table S2), while `Causalcall` has a substantially lower ratio for the human genome. This mainly indicates that unaligned bases across basecallers are mainly shared within the mapped reads, resulting in a similar number of mapped reads with similar average lengths as well as the ratio. We conclude that `RUBICALL` reads provides the highest-quality read mapping results with the largest number of mapped bases and mapped reads.

**SkipClip analysis**

Figure 9 shows the effect of `SkipClip` on validation accuracy using three different strides at which we remove a skip connection from a block, i.e., the epoch interval at which `SkipClip` removes a skip connection from a block. We use our `QABAS`-designed model that has five blocks of skip connections. We highlight the number of epochs needed to remove all the skip connections for different strides. For example, `Stride 1` requires five epochs to remove all the skip connections, while `Stride 3` requires 15 epochs. We make three observations. First, `Stride 1` converges faster to the baseline accuracy compared to `Stride 2` and `Stride 3`. By using `Stride 1`, we quickly remove all the skip connections (in five epochs) giving enough fine-tuning iterations for the model to recover its loss in accuracy. Second, all the strides show the maximum drop in accuracy (1.27−2.88%) when removing skip connections from block 1 and block 4. We observe these blocks consist of the highest number of neural network model parameters due to the skip connections (30.73% and 25.62% of the total model parameters are present in skip connections in block 1 and block 4, respectively). Therefore, the model

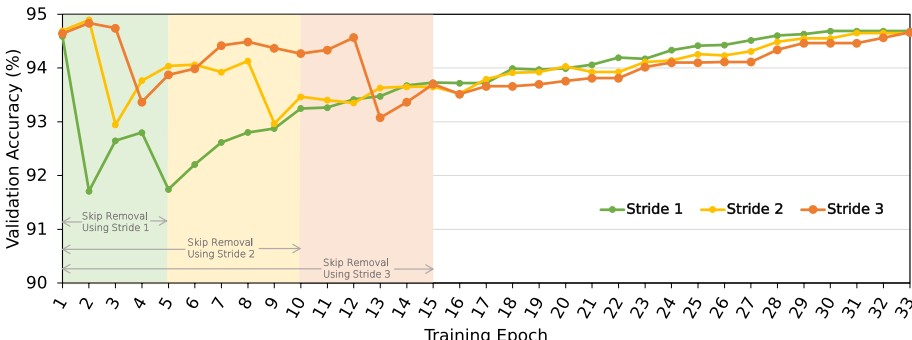

**Fig. 9** Effect of different strides while removing skip connections

requires more training epochs to recover its accuracy after the removal of skip connections from these blocks. Third, a lower stride can get rid of skip connections faster than using a higher stride. However, all strides eventually converge to the baseline accuracy at the expense of more training iterations. We conclude that `SkipClip` provides an efficient mechanism to remove hardware-unfriendly skip connections without any loss in basecalling accuracy.

### Effect of pruning RUBICALL

Figure 10 shows the effect of pruning `RUBICALL` using two different pruning methods: unstructured element pruning and structured channel pruning.

We make four major observations. First, we can remove up to 15% and 5% of model parameters providing 1.18% and 1.05% reductions in model size without any loss in accuracy by using unstructured pruning and structured pruning, respectively. However, unstructured pruning is unsuitable for hardware acceleration due to irregular structure, and structured pruning provides minimal model size (or parameters) savings. Therefore, we do not apply these pruning techniques to optimize `RUBICALL` further. Second, we observe a drop in accuracy for pruning levels greater than 15% and 5% for unstructured and structured pruning, respectively. This shows that `QABAS` found an optimal architecture as there is little room for pruning `RUBICALL` further without loss in accuracy.

Third, we observe that the *knee point* for unstructured pruning and structured pruning lies at 90% and 50%, where we achieve 80.65% and 70.10% of accuracy with 9.99× and 1.99× savings model size, respectively. After the knee point, we observe a sharp decline in accuracy. Fourth, below the knee point, we can trade accuracy for speed to further accelerate `RUBICALL` for hardware computation and resources by removing unimportant network weights. We conclude that pruning provides a tradeoff between accuracy and model size that can lead to further reductions in processing and memory demands for `RUBICALL`, depending on the type of device on which genomic analyses would be performed.

### Explainability into QABAS results

We perform an explainability analysis to understand our results further and explain `QABAS`'s decisions. The search performed by `QABAS` provides insight into whether `QABAS` has learned meaningful representations in basecalling. In Fig. 11a and b, we extract the number of model parameters and precision of each parameter in a neural network layer to calculate the total size for each layer for `Bonito_CTC` and `RUBI-CALL-MP`, respectively. We highlight each layer's precision (i.e., weights and activation

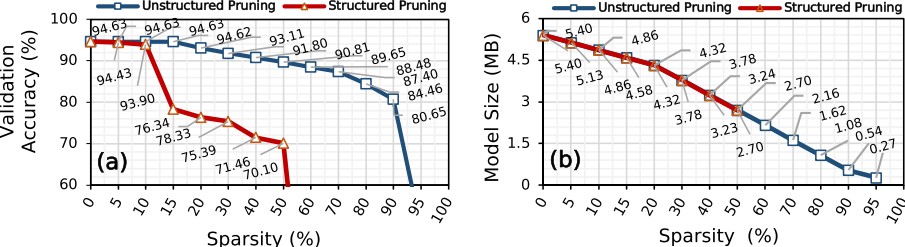

**Fig. 10** Effect of pruning `RUBICALL` on: (**a**) validation accuracy and (**b**) model size

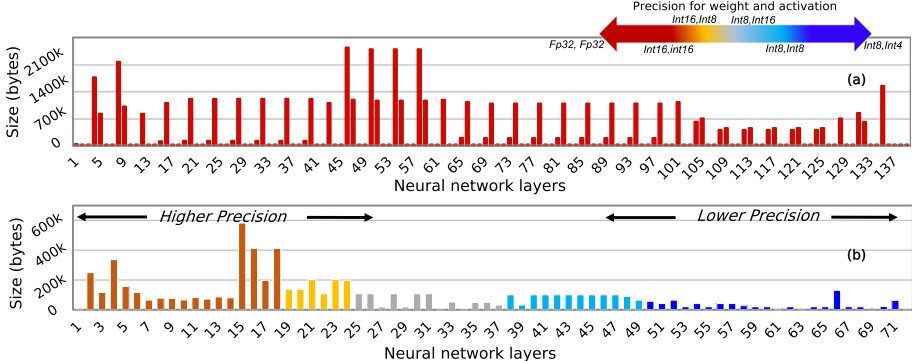

**Fig. 11** Layer size comparison for basecallers: **a** `Bonito_CTC` and **b** `RUBICALL-MP`

precision) using distinct colors. Our range includes floating-point (i.e., `fp32`) computation to integer computation (i.e., `int16`, `int8`, and `int4`) for weight and activation. Based on our experiments in the "Effect of quantization" section, we restrict the precision of weight and activation in `RUBICALL-MP` architecture in QABAS to `int8` and `int4`, respectively. We compare `RUBICALL-MP` to `Bonito_CTC` as it has the same backend (i.e., Quartznet [49]) and is designed by ONT experts. We make three observations. First, QABAS uses more bits in the initial layers than the final layers in `RUBICALL-MP`. QABAS learns that the input to RUBICALL uses an analog squiggle that requires higher precision, while the output is only the nucleotide bases (A, C, G, T), which can be represented using lower precision.

Second, RUBICALL uses 1.97× less number of neural network layers than `Bonito_CTC` while providing similar or higher basecalling accuracy on the evaluated species ("Basecalling accuracy" section). Thus, the superior performance of a basecaller architecture is not explicitly linked to its model complexity, and QABAS-designed models are parameter efficient. Third, `Bonito_CTC` uses the same single-precision floating-point representation (FP32) for all neural network layers, which leads to very high memory bandwidth and processing demands, whereas RUBICALL has every layer quantized to a different quantization domain. We conclude that QABAS provides an efficient automated method for designing more efficient and hardware-friendly genomic basecallers compared to expert-designed basecallers.

## Discussion

We are witnessing a tremendous transformation in high-throughput sequencing to significantly advance omics and other life sciences. The bioinformatics community has developed a multitude of software tools to leverage increasingly large and complex sequencing datasets. Deep learning models have been especially powerful in modeling basecalling.

### Importance of basecalling

Basecalling is the most fundamental computational step in the high-throughput sequencing pipeline. It is a critical problem in the field of genomics, and it has a significant impact on downstream analyses, such as variant calling and genome assembly. Improving the efficiency of basecalling has the potential to reduce the cost and time

required for genomic analyses, which has practical implications for real-world applications. RUBICALL offers a valuable alternative for researchers and practitioners who seek a balance between accuracy and speed. By maintaining competitive accuracy levels while significantly improving speed, our framework addresses the needs of various applications with stringent time constraints, ultimately benefiting a broader range of users. We believe that RUBICON provides a significant improvement over existing methods, and it has practical implications for the genomics community.

### Need to improve the throughput of basecallers

Increasing throughput and reducing model size is critical because of the following three reasons. First, current basecallers already have high accuracy, but biologists do not pay attention to the throughput implications of using large deep learning-based models [30]. We observe researchers building larger and larger basecallers in an attempt to gain more accuracy without heeding to the disproportionately higher amount of power these basecallers are consuming. Moreover, none of the previous basecallers [28, 29, 39–42, 44, 72] have been optimized for mixed-precision execution to reduce energy consumption. As energy usage is proportional to the size of the network, energy-efficient basecalling is essential to enable the adoption of more and more sophisticated basecallers. Second, speed is critical in certain applications and use cases, particularly those that require real-time or near-real-time processing. RUBICON addresses these needs by focusing on hardware optimization and efficient implementation, ultimately enabling faster basecalling and potentially opening up new possibilities for applications with stringent time constraints. Third, as deep learning techniques and hardware continue to evolve, the balance between accuracy and speed/energy will remain an important aspect of model development. RUBICON provides a foundation for future research and innovation in hardware-friendly deep learning models for genomic basecalling.

### Evaluating RUBICON on other platforms

All the state-of-the-art basecallers and RUBICON use high-level libraries, such as PyTorch or TensorFlow, which abstract the hardware architecture and provide a unified interface for deep learning computations. These libraries work out-of-the-box for AMD GPUs and are equally optimized for them. Currently, high-level libraries do not provide capabilities to exploit low-precision tensor cores available on the latest GPUs. As a result, existing basecallers take advantage of comparable architectural capabilities regardless of the specific GPU employed. Therefore, the hardware and software optimizations are at the same level for all supported GPU-based platforms.

### Automating basecaller generation process

Modern basecallers generally employ convolution neural networks to extract features from raw genomic sequences. However, designing a basecaller comes with a cost that a neural network model can have many different computational elements making the neural network tuning a major problem. At present, the vast majority of deep learning-based basecallers are manually tuned by computational biologists through manual trial and error, which is time-consuming. To a large extent, basecallers are being designed to provide higher accuracy without considering the compute demands of such networks. Such

an approach leads to computationally complex basecallers that impose a substantial barrier to performing end-to-end time-sensitive genomic analyses. This vast dependence of computational biologists and biomedical researchers on these deep learning-based models creates a critical need to find efficient basecalling architectures optimized for performance.

During our evaluation, we ran QABAS for 96 GPU hours to sample architectures from our search space. Using complete sampling to evaluate all the $1.8{\times}10^{32}$ viable options would take at least $\sim4.3{\times}10^{33}$ GPU hours. Thus, QABAS accelerates the basecaller architecture search to develop high-performance basecalling architectures. The final model architecture can be further fine-tuned for other hyperparameters [73, 74], such as learning rate and batch size (for example, with grid search or neural architecture search). Throughout our experiments, we build general-purpose basecalling models by training and testing the model using an official, open-source ONT dataset that consists of a mix of different species. We did not specialize basecalling models for a specific specie. Past works, such as [28], show that higher basecalling accuracy can be achieved by building species-specific models.

**Extending RUBICON**

RUBICON's modular design allows for the incorporation of additional layers or techniques, such as RNN, LSTM, and Transformers, to potentially increase accuracy further. We focus on convolution-based networks because (a) matrix multiplication is the fundamental operation in such networks that is easily amenable to hardware acceleration; (b) the training and inference of RNN and LSTM models inherently involve sequential computation tasks, which poses a challenge for their acceleration on contemporary hardware such as GPUs and field-programmable gate arrays (FPGAs) [75–83]; and (c) transformer-based models are typically composed of multiple fully connected layers, which can be supported in RUBICON by modifying convolutional layers for improved computational efficiency and performance [84]. As future work, QABAS can be extended in two ways: (1) evaluate advance model architectures (such as RNN, transformer) and (2) perform more fine-grain quantization. First, extending QABAS to other model architectures is important for researchers to quickly evaluate different computational elements. As the field of machine learning is rapidly evolving, it is non-trivial for researchers to adapt their models with the latest deep learning techniques. Second, currently, we perform mixed precision quantization, where every layer is quantized to a different domain. In the future, we can quantize every dimension of the weights to different precision. Such an approach would increase the design space of neural network architectural options to many folds. QABAS enables easy integration to explore such options automatically. Thus, QABAS is easily extensible and alleviates the designer's burden in exploring and finding sophisticated basecallers for different hardware configurations. We would explore two future directions for pruning a basecaller. First, currently, we perform one-shot pruning, whereby we prune the model once and then fine-tune the model until convergence. Another approach could be to perform iterative pruning, where after every training epoch, we can re-prune the model using certain pruning criteria. Such an approach

would further evaluate the fine-grained pruning limit of a basecaller. Second, an interesting future direction would be to combine multiple pruning techniques, e.g., structured channel pruning with structured group pruning (where we maintain the structure of the tensors without causing sparsity). Such an approach could lead to higher pruning ratios without substantial accuracy loss.

### Importance of RUBICALL beyond basecalling

For `SkipClip`, we demonstrate its applicability on basecalling only, while there are other genome sequencing tasks where deep learning models with skip connections are actively being developed, such as predicting the effect of genetic variations [72, 85], detecting replication dynamics [86], and predicting super-enhancers [87]. In Additional file 1: Section S1, we show the effect of manual skip removal, where we manually remove all the skip connections at once. We observe that the basecaller achieves 90.55% accuracy (4.08% lower than the baseline model with skip connections). By manual skip removal, the basecaller is unable to recover the loss in accuracy because CNN-based basecallers are sensitive to skip connections. Therefore, `SkipClip` provides a mechanism to develop hardware-friendly deep learning models for other genomic tasks.

### Separation between QABAS and SkipClip

Both `QABAS` and `SkipClip` share the overarching objective of creating a compact basecalling network without compromising accuracy. However, they approach this goal from distinct perspectives and employ different optimization tools. The following three points justify the separation of the two methods. First, skip connections are integral to stable model training, and by retaining them during the initial `QABAS` phase, we ensure effective training of the final basecalling network. The subsequent application of `SkipClip` allows for the controlled removal of skip connections, contributing to a more robust solution. Second, `QABAS` might find an architecture with skip connections, whereas `SkipClip` employs knowledge distillation for skip connection removal, addressing a specific aspect not efficiently handled by `QABAS` alone. Third, unlike `SkipClip`, `QABAS` tailors the neural network architecture for hardware efficiency without relying on a teacher network. The teacher network provides an upper bound on the achievable accuracy. Therefore, this two-step approach optimally combines the strengths of NAS and knowledge distillation, ensuring a comprehensive and effective optimization process for a compact and efficient basecalling model.

### Conclusion

Nanopore sequencing generates noisy electrical signals that require conversion into a standard DNA nucleotide base string through a computational process known as basecalling. Efficient basecalling is crucial for subsequent genome analysis steps. Current basecalling approaches often neglect computational efficiency, resulting in slow, inefficient, and resource-intensive basecallers. To address this, we present `RUBICON`, a framework designed for creating hardware-optimized basecallers. `RUBICON` introduces two novel machine-learning techniques: `QABAS`, an automatic architecture search for computation blocks and optimal bit-width precision, and `SkipClip`, a dynamic skip connection removal module that significantly reduces resource and storage requirements

without sacrificing basecalling accuracy. We demonstrate the capabilities of `QABAS` and `SkipClip` by designing `RUBICALL`, the first hardware-optimized basecaller, demonstrates fast, accurate, and efficient basecalling, achieving ~6.88× reductions in model size with 2.94× fewer neural network parameters compared to an expert designed basecaller. We believe our open-source implementations of `RUBICON` will inspire advancements in genomics and omics research and development.

## Methods

### Evaluation setup

Table 2 provides our system details. We evaluate `RUBICALL` using (1) AMD MI210 GPU [68] (`RUBICALL-FP`) using floating-point precision computation and (2) Versal ACAP VC2802 [64], a cutting-edge spatial vector computing system from AMD-Xilinx (`RUBICALL-MP`) using mixed-precision computation. The Versal ACAP VC2802 features Versal AI Engine ML (AIE-ML) [64] with 304 cores. The AIE-ML vector datapath implements two-dimensional single instruction, multiple data (SIMD) [88] operations using precisions ranging from int4×int8 to int16×int16 operands that can execute 512 to 64 multiply-accumulate operations (MACs) per cycle, respectively. With its many different datatype precision options, AIE-ML acts as a suitable platform to demonstrate the benefits of a mixed precision basecaller. We train all the basecallers (`Causalcall`, `Bonito_CRF-fast`, `Bonito_CTC`, `SACall`, and `Dorado-fast`) using the same MI50 GPU. We use ONNX (Open Neural Network Exchange) [89] representation to evaluate the performance on AIE-ML by calculating bit operations (BOPs) [90], which measures the number of bitwise operations in a given network, taking into account the total number of supported operations per datatype on AIE-ML.

### QABAS setup details

We use the publicly available ONT dataset [62] sequenced using MinION Flow Cell (R9.4.1) for the training and validation during the `QABAS` search phase. The dataset comprises 1,221,470 reads, all sequenced from complete genomes. This ONT training dataset has an approximate list of 496 unique taxonomic IDs using the Kraken2 [100] taxonomic classification system [101]. We randomly select 30k samples from the training set for the search phase (specified using the −chunks parameter). We use nni [102] with nn-meter [103] to implement hardware-aware NAS. We use the Brevitas library [104] to

**Table 2** System parameters and hardware configuration for the CPU, GPU and the AMD-Xilinx Versal ACAP

| | |
|---|---|
| **CPU** | AMD EPYC 7742 [91] |
| | @2.25GHz, 4-way SMT [92] |
| **Cache-Hierarchy** | 32×32 KiB L1-I/D, 512 KiB L2, 256 MiB L3 |
| **System memory** | 4×32GiB RDIMM DDR4 2666 MHz [93] PCIe 4.0 ×128 |
| **OS details** | Ubuntu 21.04 Hirsute Hippo [94], GNU Compiler Collection (GCC) version 10.3.0 [95] |
| **GPU** | AMD Radeon Instinct™ MI210 [68] 6656 Stream Processors@1.7GHz 64GB HBM2 PCIe 4.0 ×16, ROCm version 5.1.1 [96] NVIDIA A40 [69] 10,752 CUDA Cores@1.2GHz, 48GiB DRAM NVIDIA System Management Interface (NVIDIA-SMI) version 510.47.03 [97] NVIDIA CUDA Compiler Driver (NVCC) version 11.4 [98] |
| **AMD-Xilinx Versal ACAP** | Versal ACAP VC2802 [64], 304×AIE-ML@1GHz, 19MB local memory, Dual-Core Arm Cortex-A72 [99] |

perform quantization-aware training. The architectural parameters and network weights are updated using AdamW [105] optimizer with a learning rate of $2e^{-3}$, a beta value of 0.999, a weight decay of 0.01, and an epsilon of $1e^{-8}$. We set the hyperparameter $\lambda$ to 0.6. We choose these values based on our empirical analysis. After the `QABAS` search phase, the sampled networks are trained until convergence with knowledge distillation using the same ONT dataset that we use during the `QABAS` search phase, with a batch size of 64, based on the maximum memory capacity of our evaluated Mi50 GPU. We set knowledge distillation hyperparameters alpha ($\alpha$) and temperature ($\tau$) at 0.9 and 2, respectively.

### QABAS search space
For the computations operations, we search for a design with one-dimensional (1D) convolution with ten different options: kernel size (KS) options (3, 5, 7, 9, 25, 31, 55, 75, 115, and 123) for grouped 1-D convolutions. We also use an identity operator that, in effect, removes a layer to get a shallower network. For quantization bits, we use bit-widths that are a factor of $2^n$, where $2 < n < 4$ (since we need at least 2 bits to represent nucleotides A, C, G, T and 1 additional bit to represent an undefined character in case of a misprediction). We use four different quantization options for weights and activations (`<8,4>`, `<8,8>`, `<16,8>`, and `<16,16>`). We choose these quantization levels based on the precision support provided by our evaluated hardware and the effect of quantization on basecalling (see the "Discussion" section). We use five different channel sizes with four repeats each. We choose the number of repeats based on the maximum memory capacity of our evaluated GPU. In total, we have $\sim 1.8 \times 10^{32}$ distinct model options in our search space $\mathcal{M}$.

### SkipClip details
We use `Bonito_CTC` as the teacher network, while the `QABAS`-designed model is the student network. We remove skip connections with a stride 1 (using parameter `−skip_stride`). Based on hyper-parameter tuning experiments (Additional file 1: Section S2), set knowledge distillation hyperparameters alpha ($\alpha$) and temperature ($\tau$) at 0.9 and 2, respectively. We use Kullback-Leibler divergence loss to calculate the loss [106].

### Pruning details
We use PyTorch [107] modules for both unstructured and structured pruning [108] with L1-norm, i.e., prune the weights that have the smallest absolute values. We apply one-shot pruning, where we first prune a model with a specific amount of sparsity, then train the model until convergence on the full ONT dataset [62].

### Baseline basecallers
`RUBICALL` is a pure convolution-based network. We focus on convolution-based networks because (a) matrix multiplication is the fundamental operation in such networks that is easily amenable to hardware acceleration; (b) the training and inference of RNN and LSTM models inherently involve sequential computation tasks, which poses a challenge for their acceleration on contemporary hardware such as GPUs

and field-programmable gate arrays (FPGAs) [75]; and (c) transformer-based models are typically composed of multiple fully connected layers, which can be supported in `RUBICON` by modifying convolutional layers for improved computational efficiency and performance [84]. We compare `RUBICALL` against five different basecallers: (1) `Causalcall` [38] is a state-of-the-art basecaller with skip connections, (2) `Bonito_CRF-fast` [62] v0.6.2 is a recurrent neural network (RNN)-based version of basecaller from ONT that is optimized for throughput for real-time basecalling on Nanopore devices, (3) `Bonito_CTC` [62] v0.6.2 is convolutional neural network (CNN)-based hand-tuned basecaller from ONT, (4) `SACall` [43] is a transformer-based basecaller that uses an attention mechanism for basecalling, and (5) `Dorado-fast` [109] v0.4.0 is a LibTorch [110] version of `Bonito_CRF-fast` from ONT. `Dorado-fast` uses the same model architecture as `Bonito_CRF-fast` and uses the `Bonito` framework for model training. `Causalcall` and `Bonito_CTC` uses the same backend structure as `RUBICALL` (i.e., Quartznet [49]). We are aware of other basecallers such as `Halcyon` [42], `Helix` [40], and `Fast-bonito` [41]. However, these basecallers are either not open-source or do not provide training code with support for specific read formats.

### Basecalling reads

To evaluate basecalling performance, we use a set of reads generated using a MinION R9.4.1 flowcell. We use only R9 chemistry datasets as, currently, ONT does not provide a suitable public training dataset for R10 chemistry. They offer in-house trained R10 models that cannot be employed for a consistent evaluation across all basecallers. R9 and R10 chemistries involve distinct generations of nanopore technologies, including different pore proteins and read lengths. Therefore, models trained on R9 chemistry are incompatible for inference on R10 sequenced datasets. Due to these technical constraints, our study is currently limited to utilizing the available R9 chemistry training dataset from ONT and conducting inference exclusively on R9 chemistry datasets. Table 3 provides details on different organisms used in our evaluation. We use several bacterial species and the human genome. For Human HG002, we use 3× depth of coverage.

**Table 3** Details of datasets used in evaluation

| Organism | Chemistry | # Reads | Reference Genome Size (bp) |
|---|---|---|---|
| *Acinetobacter pittii* 16-377-0801 | R9.4.1 | 4467 | 3,814,719 |
| *Haemophilus haemolyticus* M1C132_1 | R9.4 | 8669 | 2,042,591 |
| *Klebsiella pneumoniae* INF032 | R9.4 | 15,154 | 5,111,537 |
| *Klebsiella pneumoniae* INF042 | R9.4 | 11,278 | 5,337,491 |
| *Klebsiella pneumoniae* KSB2_1B | R9.4 | 15,178 | 5,228,889 |
| *Klebsiella pneumoniae* NUH29 | R9.4 | 11,047 | 5,134,281 |
| *Serratia marcescens* 17-147-1671 | R9.4.1 | 16,847 | 5,517,578 |
| *Staphylococcus aureus* CAS38_02 | R9.4.1 | 16,742 | 2,902,076 |
| *Stenotrophomonas maltophilia* 17_G_0092_Kos | R9.4 | 16,010 | 4,802,733 |
| Human HG002 | R9.4.1 | 300,000 | 2,947,743,500 |

Prior to basecalling, raw nanopore signals undergo a preprocessing pipeline to prepare them for input into the neural network. Raw nanopore signals, which can be hundreds of thousands of data points long, are normalized to ensure consistent input characteristics for the subsequent processing steps. We use empirically determined normalization scaling factors from ONT's `Bonito_CTC` basecaller. The normalized signals are chunked into smaller segments, typically with overlapping regions. The chunk size and overlap are empirically set to 4000 bps and 500, respectively. Chunk size affects the balance between processing speed and accuracy. Smaller chunk sizes can lead to more accurate basecalling but may require more computational resources and time. Larger chunk sizes may be faster but can potentially introduce errors if the signal varies significantly within the chunk. Overlap represents the degree to which consecutive chunks share data with each other. Overlapping chunks can help mitigate the potential issues caused by abrupt changes in the signal at chunk boundaries. It allows for a smoother transition between chunks, reducing the chances of missing important information in the signal. However, a larger overlap may increase computational demands and processing time. After basecalling, the basecalled sequences obtained from individual signal segments are stitched back together to reconstruct the entire nucleotide sequence. The stitched sequences are then decoded to obtain the final basecalled sequences. We use the beam-search decoding [111] method to obtain the final basecalled sequences from stitched segments.

**Basecaller evaluation metrics**

We evaluate the performance of RUBICALL using two different metrics: (1) basecalling throughput (kbp/sec), i.e., the throughput of a basecaller in terms of kilo basepairs generated per second, and (2) basecalling accuracy (%), i.e., the total number of bases of a read that are exactly matched to the bases of the reference genome divided by the total length of its alignment including insertions and deletions. We measure the basecalling throughput for the end-to-end basecalling calculations, including reading FAST5 files and writing out FASTQ or FASTA file using Linux */usr/bin/time -v* command. For basecalling accuracy, we align each basecalled read to its corresponding reference genome of the same species using the state-of-the-art read mapper, minimap2 [112]. We use Rebaler [113] to generate a consensus sequence from each basecalled read set, which replaces portions of the reference genome with read-derived sequences. The assembled genome is then polished with multiple rounds of Racon [114]. This results in an assembled genome that accurately represents the original data while minimizing potential errors introduced by the reference.

**Downstream analysis**

We evaluate the effect of using RUBICALL and other baseline basecallers on two widely used downstream analyses, de novo assembly [115] and read mapping [116].

***De novo assembly***

We construct de novo assemblies from the basecalled reads and calculate the statistics related to the accuracy, completeness, and contiguity of these assemblies. To generate de novo assemblies, we use minimap2 [112] to report all read overlaps and miniasm [47] to construct the assembly from these overlaps. We use miniasm because it allows us to observe the effect of the reads on the assemblies without performing additional

error correction steps on input reads [117] and their final assembly [34]. To measure the assembly accuracy, we use dnadiff [118] to evaluate (1) the portion of the reference genome that can align to a given assembly (i.e., Genome Fraction), (2) the average identity of assemblies (i.e., Average Identity) when compared to their respective reference genomes, and (3) insertions and deletions of nucleotides (or bases) in the sequence when compared to a reference or other sequences. ( i.e., Total Indels and Indel Ratio (%)). Total indels represents the sum of all the insertions and deletions in the assembled sequence when compared to a reference or other sequences. The indel ratio is a measure of the relative abundance of indels compared to the total length of the assembled sequence (calculated using total indels/assembly length) $\times$ 100. This metric helps to understand the proportion of the assembly that contains insertions and deletions. To measure statistics related to the contiguity and completeness of the assemblies, such as the overall assembly length, average GC content (i.e., the ratio of G and C bases in an assembly), and NG50 statistics (i.e., shortest contig at the half of the overall reference genome length), we use QUAST [119]. We assume that the reference genomes are high-quality representative of the sequenced samples that we basecall the reads from when comparing assemblies to their corresponding reference genomes. The higher the values of the average identity, genome fraction, and NG50 results, the higher the quality of the assembly and, hence, the better the corresponding basecaller. When the values of the average GC and assembly length results are closer to that of the corresponding reference genome, the better the assembly and the corresponding basecaller. We use Inspector [120] to calculate the overall quality value (QV) of an assembly. The QV score is determined by considering structural and small-scale errors in proportion to the total number of base pairs in the assemblies. High-quality sequences have higher QV scores, indicating a low probability of sequencing errors, while low-quality sequences have lower QV scores, suggesting a higher likelihood of errors.

### Read mapping

We basecall the raw electrical signals into reads using each of the subject basecallers. We map the resulting read set to the reference genome of the same species using the state-of-the-art read mapper, minimap2 [112]. We use the default parameter values for mapping ONT reads using the preset parameter *-x map-ont*. We use the *stats* tool from the SAMtools library [121] to obtain four key statistics on the quality of read mapping results, the total number of mismatches, the total number of mapped bases, the total number of mapped reads, and the total number of unmapped reads. We normalize the total number of base mismatches and the total number of mapped bases using the total number of bases in the reads, while for the total number of mapped reads and the total number of unmapped reads, we normalize using the total number of reads.

## Supplementary Information

**Additional file 1.** Supplementary notes S1-S6, Figs. S1-S9, and Tables S1-S3.

**Additional file 2.** Review history.

**Acknowledgements**
We thank the SAFARI Research Group members for their valuable feedback and the stimulating intellectual and scholarly environment they provide. SAFARI Research Group acknowledges the generous gifts of their industrial partners, including Google, Huawei, Intel, Microsoft, VMware, and AMD. This research was partially supported by the Semiconductor Research Corporation. SAFARI Research Group acknowledges support from the European Union's Horizon program for research and innovation under grant agreement No. 101047160, project BioPIM. Special thanks to Alessandro Pappalardo for his support with quantization-aware training. We appreciate valuable discussions with Giovanni Mariani. Thanks to AMD for providing access to the HPC fund cluster [122].

**Peer review information**

**Review history**
The review history is available as Additional file 2.

**Authors' contributions**
G.S., M.A., K.D., and A.K. conceived RUBICON. G.S. designed and implemented RUBICON. G.S., C.F., and M.C. collected data and performed the evaluations. K.D., H.C., and O.M. supervised the work. G.S., M.A., K.D., and C.F. wrote the manuscript. All authors reviewed and edited the manuscript. All authors analyzed the results. All authors read and approved the final manuscript.

**Funding**

**Availability of data and materials**
The read set and reference set used in this study are part of work carried out by Wick et al. [28], which can be downloaded from https://bridges.monash.edu/articles/dataset/Raw_fast5s/7676174 and https://bridges.monash.edu/articles/dataset/Reference_genomes/7676135, respectively. For the human genome [123], we download reads from https://labs.epi2me.io/gm24385_2020.11/, while the reference genome is available at https://github.com/marbl/HG002. All trained models and generated reads can be downloaded from https://zenodo.org/record/10198815. We ensure unbiased, fair, and consistent evaluation by retraining all the basecallers using the official ONT dataset [62].
Source code with the instructions for reproducing the results is publicly available at GitHub [124] and Zenodo [125]. Scripts used to perform basecalling accuracy analysis are available at: https://github.com/rrwick/Basecalling-comparison.

# Declarations

**Ethics approval and consent to participate**
Not applicable—ethical approval was not needed for the study, as publicly available datasets were used. No private, confidential, or sensitive information pertaining to individuals was utilized. Furthermore, our research did not involve any animal experiments.

**Competing interests**
Gagandeep Singh, Kristof Denolf, and Alireza Khodamoradi are affiliated with AMD. The remaining authors declare no competing interests.

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

## 