## [**Additional file 2.** Review history. · Genome Biology]

Review History

First round of review

Reviewer 1

Were you able to assess all statistics in the manuscript, including the appropriateness of statistical tests used? Yes, and I have assessed the statistics in my report.

Were you able to directly test the methods? No.

Comments to author:

Singh et al present RUBICALL a thoroughly benchmarked alternative nanopore basecaller that is optimized for speed. It performs on par with current state of the art, while being orders of magnitude better. The results are convincing and well-presented (albeit a bit verbose and lengthy). Some considerations for improvement are given below.

Comment 1:

Basecalling is indeed a very compute intensive step that can take several hours on large datasets. The reviewers motivate their research to improve basecalling speed by simply presenting basecalling as a bottleneck step. I would argue that in most cases, this step is done during sequencing, which results in no additional wait time by the scientist. And for these cases, a more efficient basecalling time would only translate to a reduction in power consumption by the hardware.

There are however cases where having efficient basecallers can have a positive impact, here a few examples:

- Re-basecalling of already existing datasets with higher accuracy models.
- Field sequencing, where high compute hardware is not available.
- Adaptive sampling, where one must quickly basecall the data so that it can rapidly be aligned to make a decision whether to keep sequencing or eject the read.

I would suggest the authors to further motivate in which cases having a faster or more efficient basecaller would make an impact and perhaps include a cost (saving) calculation. There is some motivation in the discussion, but I believe this motivation should also be made in the introduction.

Comment 2

The authors state: For example, Bonito_CRF-fast, a throughput-optimized recurrent neural network-based version of Bonito [45] basecaller from ONT, takes ~6 hours to basecall a 3 Gbps (Giga base pairs) human genome on a powerful server-grade GPU. I am a bit surprised by how slow that is, since in this published benchmark by ONT (<https://aws.amazon.com/blogs/hpc/benchmarking-the-oxford-nanopore-technologies-basecallers-on-aws/>) they show that it can take at the slowest 62 hours to basecall 90 Gbps of

data, which would translate that you would need at max 2 hours (probably less since the benchmark uses the high accuracy model, not the fast one) to basecall 3 Gbps. This is with a reasonable hardware setup: 1GPU with 16Gb VRAM, 16Gb RAM and 4 CPUs. I would encourage the authors to revise why there are such discrepancies between their results and the ones from ONT.

Comment 3

There is a lack of information on the methods section regarding the processing of the data before and after basecalling. Raw nanopore signals can be hundreds of thousands long which makes it not feasible to use as input directly to the neural network. The nanopore raw signal is normalized (not discussed in the manuscript), chunked (not discussed in the manuscript) into small pieces (usually overlapping) and then fed into the basecaller. Then, the output of the neural network is stitched back together (not discussed in the manuscript) and decoded (not discussed in the manuscript). Some parameters, like chunk size, overlap and decoding method (greedy, beam-search) are completely omitted in the methods.

Comment 4:

The introduction is quite verbose. I certainly see that this intro would be necessary for the average GB reader, as many computational biologists will be unfamiliar with many of these concepts. However, I believe some of it should either be shortened or moved to the methods, to speed up the transition to the results. For example, SkipClip requires 4 paragraphs to describe, which in my opinion, is the simplest of the approaches to explain. The introduction also uses 4 figure panels, I would suggest having a single panel that quickly summarizes the different approaches, and provide more detailed figures in the supplementary material.

Comment 5:

Although the main goal of the manuscript is the development of accurate and very fast basecallers; and for this reason the comparison of RUBICALL is to also fast basecallers. Despite this, it would be good to have an idea of what would be the "maximum" achievable performance. This is done on Figure S3, where Bonito SUP is included; however, the performance of Bonito SUP is not mentioned anywhere in the main text, which is a missed opportunity.

Comment 6:

I might have missed something, but it is not entirely clear why for some analysis only RUBICALL is compared against the other basecallers, and not RUBICALL-FP and RUBICALL-MP. For example Fig9 has both, but Fig10 has only RUBICALL. It would be best to have both in all figures. If there's a good reason for that not to happen, then properly annotate if RUBICALL means RUBICALL-MP or RUBICALL-FP.

Minor comments

- References to Figure 1 and 2 use numeric systems, instead of the traditional letter system. References should read (Fig 1a, Fig1b, etc.)

- Figure 13 and figure 15 need text explaining the figure.
- Figure S1 denotes the change in number of parameters in gray inside the plot frame box. I find it a bit confusing since it seems that the baseline model has 94.63 parameters, but that value is the accuracy. I suggest to move the #parameters as well as the values outside the plot frame box.
- Figure S2 I assume also talks about validation accuracy in Bonito_CTC architecture, might be best to also explicitly mention that.
- Figure S3c, seems a bit counter-intuitive to have basecalling accuracy axis flipped.
- Table S1, the basecalling accuracy metrics for RUBICALL-FP and RUBICALL-MP are exactly the same, this might be true, but perhaps double-check this is not a copy-paste error.
- Figure numbers are incorrect: there is no figure 6 nor 11.
- Table 1 has some values in bold, the meaning of these (I assume it's best performing) is not explained in the table text.
- Table 1, for assembly length I assume the value closest to the real assembly length (based on the reference) is best. However, the bold number does not always correspond to that. For example, for Haemophilus haemolyticus M1C132_1, RUBICALL is bold, but the values of Dorado-fast and Bonito_CRF-fast are closest to the reference assembly length.
- Table 1, Klebsiella pneumonia INF032 and Serratia marcescens 17-147-1671, the assembly length and NG50 for SACall values are not formatted properly as they lack the thousands comma separators.
- Table 1, for many cases the Assembly length and NG50 values are exactly the same. This could be correct, but I would encourage the authors to re-check just in case these are not copy-paste mistakes.

Reviewer 2

Were you able to assess all statistics in the manuscript, including the appropriateness of statistical tests used? No, I do not feel adequately qualified to assess the statistics.

Were you able to directly test the methods? No.

Comments to author:

In this paper, the authors focus on developing a learning framework named RUBICON for efficient nanopore basecalling. This hardware-optimized framework entails two key steps:

- (1) Searching for the optimal network structure tailored to the specific hardware platform.
- (2) Removing skip-connections in the neural network through a process called knowledge distillation.

The authors report a roughly four-fold speed increase and approximately 3% improvement in basecalling accuracy when compared with a fast, state-of-the-art basecaller.

The following are my primary comments and questions regarding this work.

1. Generalization

The work was mainly conducted on AMD GPU and XILINX accelerator platforms, which are not commonly used in most labs, where X86+Nvidia GPUs are more commonly used at the current time point. Although the proposed method is not platform-specific, authors are expected to demonstrate the performance of their framework on other different platforms.

2. Pipeline Component

The two major components of the proposed method involve creating a compact network structure for a basecalling model while preserving the model's efficacy. Could step-2 not be resolved by step-1, considering that the skip-connection could also be treated as a model component, and step-1 aims to find the best model structure? Does this indicate that the QABAS in step-1 can only search for certain specific model architectures?

Or to say, are steps 1 and 2 addressing the same problem with different tools (NAS, distillation)?

3. Application Scenario

The proposed framework is optimized with hardware constraints in mind, but this optimization incurs a computational cost. Different end users may use different hardware. Although the optimized model can be highly efficient in the decoding stage, users are required to optimize the neural network model according to their specific hardware and then re-train the basecaller. Can the authors provide a more universal hardware setting for direct deployment? Besides being hardware-friendly, user-friendliness is also a crucial consideration.

4. Evaluation of Datasets

The study mainly evaluated nanopore R9.4 reads from nine prokaryotic species. A more comprehensive result might be achieved by using other benchmark datasets (e.g., HG002) and the latest R10.4 chemistry with publicly available datasets.

Reviewer 3

Were you able to assess all statistics in the manuscript, including the appropriateness of statistical tests used? Yes, and I have assessed the statistics in my report.

Were you able to directly test the methods? No.

Comments to author:

Singh et al present a new framework for nanopore basecalling architecture. Their work has resulted in two contributions, QABAS and SkipClip. The authors did a good job in summarizing the current state of basecalling frameworks for available models. The problem statements, with

the five reasons, are appropriate. I liked the approach taken wherein pruning is applied to trim down the models so that their performance can be optimized and sped up.

I have a few comments.

1. It might be good to explain the specifics of data being discussed in the introduction. For example, I was not sure if the authors meant basecalling 3 Gb of sequence (1x human genome) takes 6 hours using Bonito.
2. It might also be good to define performance on other GPUs which, I think, would be relevant for users. Was the performance measured on NVIDIA/ARM?
3. I would not call bonito as state-of-the-art. It was the R&D basecaller that ONT released to the community. However, the SOTA for Nanopore at the time would have been Guppy (and now a new version of Dorado). I appreciate the source code for those is not open. I do, however, feel that the performance (in runtime, speed, and quality) would benefit from being measured relative to those for users in the community.
4. Figure 8c. It is odd to see a decrease in accuracy on the axis (which makes it hard to follow).
5. It may also be worth breaking down the performance comparisons (possibly as a supplementary figure) to compare fast vs. fast and higher accuracy (e.g. Bonito-CTC, or sup) with high accuracy models.
6. It would be useful to compare under and over represented sequences (kmers) for the reads and resulting assemblies.
7. The QVs should be compared as well to ascertain that RUBICALL data based assemblies are not only covering higher percent of the genome but also better in consensus quality.
8. It would be useful to apply and measure the performance and quality for human genome (or a chromosome perhaps?).
9. If the # mapped reads and # bases mapped are comparable across the basecallers, does it suggest that other basecallers include can call unalignable sequence either within a read or (more likely) as additional reads?
10. How do indel errors compare between these basecallers?
11. What species are the training data from?
12. I did not see where the explanation and application for hardware coupling and optimization was. It is talked about in the paper, but I was unable to see the implementation or details. Could the authors please clarify?
13. I appreciate the code being shared but will say that figshare is an odd choice. Is there a plan to also share via GitHub?
14. As a suggestion, I think a Jupyter notebook might be a nice addition to the codebase.

Overall, this work will advance the work on basecallers. My best wishes to the authors.

Summary of Changes and Response to Reviewers for: A Framework for Designing Efficient Deep Learning-Based Genomic Basecallers

We thank the reviewers very much for their thoughtful reviews of our manuscript. We also thank the editor-in-chief, Kevin Pang, for the time and effort.

This document summarizes the improvements that we have made to our paper. The revision is centered around the reviewer's comments. We have addressed all comments as thoroughly as possible. We will further refine the text and improve the clarity of the paper. **Our modifications are highlighted in blue font in the revised manuscript.**

We first list the major changes we made to the paper in this revision. We then discuss common reviewer comments, followed by individual reviewer's comments, and provide a detailed explanation of the changes we have made. Within each section, each text box addresses one or a set of related comments. The silver boxes list pointers to reviewer comments and response(s) that the text box addresses in an effort to keep this manuscript self-contained and easy to follow.

MAJOR CHANGES TO THE PAPER

Based on the reviewers' feedback, the following is a list of the significant changes we have made to the paper:

1. **[Evaluation Using the Human Genome]** To underscore the applicability of our approach beyond bacterial species, we have incorporated an evaluation using the human genome HG002 in Section **2.5.2 Read Mapping**. This comprehensive assessment with a complex and highly relevant dataset highlights the practical utility of the RUBICON framework on a critical genomics workload.
2. **[Experiments on Other Platforms]** We expanded our evaluation by conducting experiments on the NVIDIA A40 GPU-based system in Supplementary Section **S5 Evaluation on Other Hardware Platforms**, ensuring the adaptability and robustness of RUBICON across various hardware environments. This broader platform assessment enhances the applicability and versatility of our approach.
3. **[Code Open-Sourcing]** We have fully open-sourced the code for RUBICON at: <https://github.com/Xilinx/neuralArchitectureReshaping>. This initiative promotes transparency, encourages community engagement, and allows researchers to relatively quickly implement and build upon our methods, fostering the field's growth.
4. **[Streamlining the Entire Text]** We have condensed and streamlined the entire manuscript to provide a more concise and reader-friendly presentation. Our revised manuscript conforms to the style for Genome Biology Methods articles. By removing unnecessary redundancy and focusing on the most critical information, we ensure that the core concepts and findings are easily accessible and comprehensible, enhancing the clarity and impact of our research.

1. Common Comments

1.1. Evaluation using human genome

[Rev. 2] The study mainly evaluated nanopore R9.4 reads from nine prokaryotic species. A more comprehensive result might be achieved by using other benchmark datasets (e.g., HG002)

[Rev. 3] It would be useful to apply and measure the performance and quality for human genome (or a chromosome perhaps?)

Thank you for your valuable feedback regarding evaluating RUBICALL on human genome data, particularly using the human HG002 dataset. We appreciate your suggestion and recognize the importance of expanding the scope of our evaluation.

In our revised version, we have added new experiments using HG002 for all the evaluated basecallers. This evaluation allows us to assess the performance and quality of our framework in a context that is of significant interest to the broader scientific community.

We updated Section 6 **Data Availability**, with details on reads and reference for the human dataset:

For the human genome, we download reads from https://labs.epi2me.io/gm24385_2020.11/, while the reference genome is available at <https://github.com/marbl/HG002>.

We also updated Section 5 **Methods** Table 3 with details on the human dataset:

Organism	Chemistry	# Reads	Reference Genome Size (bp)
Acinetobacter pittii 16-377-0801	R9.4.1	4,467	3,814,719
Haemophilus haemolyticus M1C132_1	R9.4	8,669	2,042,591
Klebsiella pneumoniae INF032	R9.4	15,154	5,111,537
Klebsiella pneumoniae INF042	R9.4	11,278	5,337,491
Klebsiella pneumoniae KSB2_1B	R9.4	15,178	5,228,889
Klebsiella pneumoniae NUH29	R9.4	11,047	5,134,281
Serratia marcescens 17-147-1671	R9.4.1	16,847	5,517,578
Staphylococcus aureus CAS38_02	R9.4.1	16,742	2,902,076
Stenotrophomonas maltophilia 17_G_0092_Kos	R9.4	16,010	4,802,733
Human HG002	R9.4.1	300,000	2,947,743,500

We updated all our results to include the human genome. We make the following four changes in Section 2 **Results**. First, Section 2.3 **Performance Comparison** includes performance results on all the evaluated basecallers with the Human HG002 dataset. We observe RUBICALL -MP is the highest performing basecaller and improves average performance by 364.89 \times , 14.25 \times , 128.13 \times , 81.58 \times , and 3.77 \times over

Causalcall, Bonito_CRF-fast, Bonito_CTC, SACall, and Dorado-fast, respectively.

Figure 6: Performance comparison of RUBICALL (using floating-point precision (RUBICALL-FP) and mixed-precision (RUBICALL-MP)) and five state-of-the-art basecallers on AMD MI210. The y-axis is on a logarithmic scale.

Second, we updated Section 2.4 **Basecalling Accuracy** to demonstrate the basecalling accuracy of all the evaluated basecallers on Human HG002.

Figure 7: Basecalling accuracy comparison of RUBICALL (using floating-point precision (RUBICALL-FP) and mixed-precision (RUBICALL-MP)).

Third, we updated Section 2.5.1 **De Novo Assembly** to show the accuracy, completeness, and contiguity of assemblies we generate using our evaluated basecaller in terms of Genome Fraction (%), Average Identity (%), Assembly Length, Average GC (%), NG50, Total Indels, Indel Ratio (%), and Quality Value (QV) in Table 1.

Table 1: Assembly quality comparison of the evaluated basecallers for different species. We measure assembly accuracy in terms of genome fraction (Genome Fraction (%)) and average identity (Average Identity (%)). Genome fraction is the portion of the Reference genome that can align to a given assembly, while average identity is the average of the identity of assemblies when compared to their respective Reference genomes. We measure statistics related to the contiguity and completeness of the assemblies in terms of the overall assembly length (Assembly Length), Average GC content (Average GC (%)) (i.e., the ratio of G and C bases in an assembly), NG50 statistics (NG50) (i.e., shortest contig at the half of the overall Reference genome length), total number of indels in all aligned bases in the assembly (Total Indels), the ratio of indels to assembly length (Indel Ratio (%)), and the reliability of basepairs using the quality value (Quality Value). NA indicates that the generated assemblies were unalignable to the reference genome.

Dataset	Basecaller	Genome Fraction (%)	Average Identity (%)	Assembly Length	Average GC (%)	NG50	Total Indels	Indel Ratio (%)	Quality Value (QV)
Acinetobacter pittii 16-377-0801	Causalcall	92.45	86.18	3,826,077	42.23	3,826,077	270,228	7.06	11.99
	Bonito_CRF-fast	96.64	89.29	3,628,317	38.82	3,628,317	242,373	6.68	12.03
	Bonito_CTC	96.87	91.44	3,676,821	38.9	3,676,821	210,496	5.72	12.45
	SACall	96.68	89.42	3,699,232	38.7	3,699,232	247,997	6.7	12.1
	Dorado-fast	96.37	88.72	3,839,847	39.09	3,839,847	245,016	6.38	12.03
	RUBICALL	96.87	91.51	3,694,086	38.82	3,694,086	208,748	5.65	15.42
Reference	100	100	3,814,719	38.78	3,814,719	0	0	-	
Haemophilus haemolyticus M1C132_1	Causalcall	0.00	0.00	0	0	0	0	0	NA
	Bonito_CRF-fast	88.76	91.51	2,046,024	37.98	2,046,024	128,481	6.28	12.25
	Bonito_CTC	96.87	90.70	1,957,480	38.87	1,957,480	118,253	6.04	15.34
	SACall	90.11	88.45	2,032,994	38.22	1,880,730	134,702	6.63	13.15
	Dorado-fast	89.42	88.97	2,110,860	39.49	2,110,860	129,503	6.14	12.38
	RUBICALL	96.87	90.54	1,966,781	38.92	1,966,781	119,777	6.09	15.37
Reference	100	100	2,042,591	38.46	2,042,591	0	0	-	
Klebsiella pneumoniae INF032	Causalcall	92.45	87.35	4,959,127	56.9	4,959,127	353,550	7.13	10.54
	Bonito_CRF-fast	92.69	87.53	4,761,297	57.19	4,761,297	347,299	7.29	10.56
	Bonito_CTC	94.50	90.20	4,897,352	56.65	4,897,352	317,428	6.48	11.26
	SACall	93.97	88.08	4,874,880	56.87	4,874,880	379,028	7.78	10.8
	Dorado-fast	93.00	87.69	5,063,562	56.8	5,063,562	348,572	6.88	10.64
	RUBICALL	94.51	90.30	4,924,240	56.85	4,924,240	314,651	6.39	11.27
Reference	100	100	5,111,537	57.63	5,111,537	0	0	-	
Klebsiella pneumoniae INF042	Causalcall	91.44	87.36	5,288,166	56.94	5,288,166	374,162	7.08	10.84
	Bonito_CRF-fast	92.08	88.49	5,052,889	56.8	5,052,889	357,354	7.07	10.93
	Bonito_CTC	93.12	90.49	5,111,083	56.61	5,111,083	317,075	6.2	11.40
	SACall	92.93	88.60	5,149,039	56.72	5,149,039	369,388	7.17	11.08
	Dorado-fast	90.21	88.20	5,737,059	56.44	5,401,717	342,141	5.96	10.98
	RUBICALL	93.12	90.60	5,146,050	56.72	5,146,050	312,448	6.07	11.42
Reference	100	100	5,337,491	57.41	5,337,491	0	0	-	
Klebsiella pneumoniae KSB2_1B	Causalcall	91.58	86.97	5,175,311	57.09	5,175,311	363,807	7.03	10.88
	Bonito_CRF-fast	90.24	88.00	4,932,626	56.71	4,932,626	357,769	7.25	10.86
	Bonito_CTC	93.07	90.11	5,003,377	56.69	5,003,377	320,519	6.41	11.41
	SACall	93.58	88.19	5,034,408	56.79	5,034,408	372,380	7.4	11.16
	Dorado-fast	90.28	87.67	5,442,186	56.72	5,261,731	349,387	6.42	11.03
	RUBICALL	93.07	89.89	5,023,639	56.75	4,932,626	357,769	7.12	11.25
Reference	100	100	5,228,889	57.59	5,228,889	0	0	-	
Klebsiella pneumoniae NUH29	Causalcall	89.08	86.01	5,158,874	56.78	5,158,874	389,676	7.55	11.75
	Bonito_CRF-fast	92.17	89.34	4,942,833	57.01	4,942,833	355,690	7.2	11.47
	Bonito_CTC	94.36	90.26	4,918,147	57.04	4,918,147	324,406	6.6	11.92
	SACall	93.66	88.58	4,978,307	57.06	4,978,307	360,950	7.25	11.56
	Dorado-fast	92.27	88.12	5,195,594	57.01	5,195,594	355,728	6.85	11.56
	RUBICALL	94.36	90.43	4,940,813	57.18	4,940,813	316,019	6.4	11.83
Reference	100	100	5,134,281	57.61	5,134,281	0	0	-	
Serratia marcescens 17-147-1671	Causalcall	89.91	86.23	5,532,953	57.86	5,422,052	401,545	7.26	13.39
	Bonito_CRF-fast	96.06	89.56	5,479,812	58.85	5,282,474	345,351	6.3	12.66
	Bonito_CTC	96.76	91.38	5,534,329	58.41	5,316,651	298,982	5.4	13
	SACall	94.29	89.36	5,366,913	58.57	5,366,913	358,954	6.69	12.27
	Dorado-fast	96.51	88.87	5,758,989	58.29	5,282,474	348,968	6.06	12.5
	RUBICALL	96.76	91.59	5,597,251	58.52	5,346,640	294,643	5.26	13.01
Reference	100	100	5,517,578	59.13	5,517,578	0	0	-	
Staphylococcus aureus CAS38_02	Causalcall	94.35	87.29	2,849,123	36.59	2,810,038	191,730	6.73	10.8
	Bonito_CRF-fast	96.27	91.49	2,790,895	33.05	2,752,169	149,623	5.36	11.59
	Bonito_CTC	97.03	93.57	2,858,986	32.86	2,819,356	123,542	4.32	12.82
	SACall	95.66	91.25	2,837,503	32.91	2,798,079	165,200	5.82	11.57
	Dorado-fast	96.70	91.16	2,927,882	33.52	2,752,169	152,216	5.2	11.64
	RUBICALL	97.03	93.36	2,860,885	33.24	2,821,276	124,795	4.36	12.59
Reference	100	100	2,902,076	32.82	2,902,076	0	0	-	
Stenotrophomonas maltophilia 17_G_0092_Kos	Causalcall	94.85	85.73	4,823,177	63.66	4,823,177	366,228	7.59	11.01
	Bonito_CRF-fast	94.60	89.74	4,596,898	65.5	4,596,898	337,040	7.33	11.10
	Bonito_CTC	95.42	90.14	4,664,226	64.82	4,664,226	298,711	6.4	11.51
	SACall	95.28	88.50	4,672,540	64.98	4,672,540	339,853	7.27	11.11
	Dorado-fast	92.99	87.70	4,854,007	63.99	4,854,007	337,105	6.94	11.01
	RUBICALL	95.46	90.49	4,693,744	65.03	4,693,744	289,073	6.16	11.63
Reference	100	100	4,802,733	66.28	4,802,733	0	0	-	
Human HG002	Causalcall	NA	NA	130,962	42.95	13,522	NA	NA	NA
	Bonito_CRF-fast	0.002	92.36	119,570,537	40.34	368,848	2,860	0	18.87
	Bonito_CTC	0.430	95.06	134,732,516	40.86	371,590	384,243	0.29	18.58
	SACall	NA	NA	63,025,520	39.87	320,873	NA	NA	NA
	Dorado-fast	0.001	93.15	121,146,376	39.8	361,677	926	0	17.46
	RUBICALL	0.125	94.50	140,928,248	40.99	393,950	100,256	0.1	17.81
Reference	100	100	2,947,743,500	40.79	2,947,743,500	0	0	-	

Fourth, we updated Section 2.5.2 Read Mapping, where we compare the total number of base mismatches, the total number of mapped bases, the total number of mapped reads, and the total number of unmapped reads.

Figure 8: Comparison of RUBICALL (using floating-point precision (RUBICALL-FP) and mixed-precision (RUBICALL-MP)) for normalized (a) mismatches, (b) bases mapped, (c) reads mapped, and (d) reads unmapped.

In our revised manuscript, we use normalized values instead of raw values for read mapping analysis to avoid the average values (AVG in Figure 8(a), 8(b), 8(c), and 8(d)) being biased toward the large human genome (i.e., HG002 has upto $67\times$ more reads than our previously evaluated bacterial species). We normalize the total number of base mismatches and the total number of mapped bases using the total number of bases in the reads, while for the total number of mapped reads and the total number of unmapped reads, we normalize using the total number of reads. We updated Section 5 **Methods** to include the above information:

We normalize the total number of base mismatches and the total number of mapped bases using the total number of bases in the reads, while for the total number of mapped reads and the total number of unmapped reads, we normalize using the total number of reads.

Based on our experiments on HG002, we add the following new observation in Section 2.5.2 **Read Mapping**:

Fifth, for each dataset, we find that the ratio of the number of mapped bases to the number of mapped reads and the average length of the reads are mainly similar across all basecallers (Supplementary Table S2), while Causalcall has a substantially lower ratio for the human genome. This mainly indicates that unaligned bases across basecallers are mainly shared within the mapped reads, resulting in a similar number of mapped reads with similar average lengths as well as the ratio.

1.2. Using NVIDIA GPUs and extending to other platforms

[Rev. 2] The work was mainly conducted on AMD GPU and XILINX accelerator platforms, which are not commonly used in most labs, where X86+Nvidia GPUs are more commonly used at the current time point. Although the proposed method is not platform-specific, authors are expected to demonstrate the performance of their framework on other different platforms.

[Rev. 3] It might also be good to define performance on other GPUs which, I think, would be relevant for users. Was the performance measured on NVIDIA/ARM?

Thank you for highlighting the importance of demonstrating the performance of our framework on a broader range of platforms, especially the commonly used NVIDIA GPUs. To address the generalization comment, we make two significant changes. First, we have added new experiments in Supplementary Section **S5 Evaluation on Other Hardware Platforms**, where we measure results on NVIDIA A40 GPU. We have also updated our hardware setup to use the newer AMD generation of AMD GPU, MI210, to have a fair comparison between the two evaluated GPUs.

S5. Evaluation on Other Hardware Platforms

We also evaluate the performance of RUBICALL and all the other basecallers on NVIDIA A40 [69] GPU with 48GiB DRAM and AMD EPYC 7442 [86] 24-Core with 256GiB DRAM. Compared to the AMD MI210 [68], the NVIDIA A40 has a $1.65\times$ higher peak compute performance while maintaining a $2.35\times$ lower peak memory bandwidth.

We make two major observations from Figure S7. First, RUBICALL-MP on AIE consistently outperforms A40 by $502.52\times$, $14.67\times$, $104.14\times$, $111.25\times$, $45.61\times$, and $3.19\times$ higher performance compared to Causalcall, Bonito_CRF-fast, Bonito_CTC, SACall, RUBICALL-FP, and Dorado-fast, respectively. Second, for compute-bound basecallers, A40 provides $1.23\times$, $1.18\times$, and $1.09\times$ higher performance than AMD MI210 (Figure 6) for Bonito_CTC, Dorado-fast, and RUBICALL-FP, respectively. For memory-bound basecallers, A40 provides $1.38\times$, $1.03\times$, and $1.36\times$ lower performance for Causalcall, Bonito_CRF-fast, and SACall, respectively. We conclude that RUBICON provides benefits across multiple hardware platforms.

Figure S7: Performance comparison of RUBICALL (using floating-point precision (RUBICALL-FP) and mixed-precision (RUBICALL-MP)) and five state-of-the-art basecallers on NVIDIA A40 [69]. The y-axis is on a logarithmic scale.

We refer to our new results in Section 2.3 **Performance Comparison** as follows:

Figure S7 in Supplementary Section S5 demonstrates the performance of all the evaluated basecallers on NVIDIA A40 [69] GPU.

Second, in our recently open-sourced RUBICON code [127], we provide installation scripts for both AMD [R1] and NVIDIA GPUs [R2].

[R1] https://github.com/Xilinx/neuralArchitectureReshaping/blob/main/requirements_rocm.txt

[R2] https://github.com/Xilinx/neuralArchitectureReshaping/blob/main/requirements_cuda.txt

We believe that our updated evaluation on a different platform provides a robust assessment of RUBICON and ensures our framework’s efficacy and efficiency across different popular hardware configurations.

1.3. Counter-intuitive accuracy axis of Figure 8c and Figure S3c

[Rev. 1] Figure S3c, seems a bit counter-intuitive to have basecalling accuracy axis flipped.

[Rev. 3] Figure 8c. It is odd to see a decrease in accuracy on the axis (which makes it hard to follow).

The basecalling accuracy axis in Figure 8c and Figure S3c has been adjusted, which are Figure 5(a) and Figure S6 in the revised manuscript, respectively.

Figure 5: Comparison of average basecalling throughput for RUBICALL-MP with state-of-the-art basecallers in terms of: (a) average basecalling accuracy, (b) model parameters, and (c) model size. RUBICALL-MP provides higher compute performance with lower model size when compared to RUBICALL-FP because of the mixed-precision computation.

Figure S6: Comparison of average basecalling throughput for RUBICALL-MP with baseline basecaller in terms of: (a) average basecalling accuracy, (b) model parameters, and (c) model size.

1.4. Motivational data

[Rev. 1] The authors state: For example, Bonito_CRF-fast, a throughput-optimized recurrent neural network-based version of Bonito [45] basecaller from ONT, takes ~6 hours to basecall a 3 Gbps (Giga base pairs) human genome on a powerful server-grade GPU. I am a bit surprised by how slow that is, since in this published benchmark by ONT (<https://aws.amazon.com/blogs/hpc/benchmarking-the-oxford-nanopore-technologies-basecallers-on-aws/>), they show that it can take at the slowest 62 hours to basecall 90 Gbps of data, which would translate that you would need at max 2 hours (probably less since the benchmark uses the high accuracy model, not the fast one) to basecall 3 Gbps. This is with a reasonable hardware setup: 1GPU with 16Gb VRAM, 16Gb RAM and 4 CPUs. I would encourage the authors to revise why there are such discrepancies between their results and the ones from ONT.

[Rev. 3] It might be good to explain the specifics of data being discussed in the introduction. For example, I was not sure if the authors meant basecalling 3 Gb of sequence (1x human genome) takes 6 hours using Bonito.

Earlier motivational numbers relied on an older version of the bonito basecaller. The benchmark results in the shared link (<https://aws.amazon.com/blogs/hpc/benchmarking-the-oxford-nanopore-technologies-basecallers-on-aws/>) are based on different basecallers from Oxford Nanopore Technologies (ONT) (closed-source Guppy and their newer throughput-optimized basecaller, Dorado). We have reviewed the benchmarking data provided by Oxford ONT and updated our motivation to better explain the specifics of the data in Section 1 **Background** accordingly.

For example, state-of-the-art throughput optimized basecaller, Dorado-fast, takes ~2.1 hours to basecall a 300 Gbps (Giga basepairs) human genome at 3x coverage on a server-grade GPU (NVIDIA A10G [46] GPU with 24GiB DRAM and 16x CPU with 64 GiB DRAM) [47], while the subsequent step, i.e., read mapping, takes only a small fraction of basecalling time (~0.11 hours using minimap2 [48]).

2. Individual Comments

Reviewer 1

[IR1.1] I would suggest the authors to further motivate in which cases having a faster or more efficient basecaller would make an impact and perhaps include a cost (saving) calculation. There is some motivation in the discussion, but I believe this motivation should also be made in the introduction.

We appreciate your insights regarding the motivation and potential impact of improving basecalling speed. We have carefully considered your suggestions and have made the following two revisions to address these concerns.

First, we have revised Section 1 **Background** to explicitly outline scenarios where a faster or more efficient basecaller would substantially impact. This includes emphasizing cases such as field sequencing and adaptive sampling, where the rapid basecalling of data is critical due to limited hardware availability or the need for real-time decision-making. We now highlight the potential benefits in various contexts, including the re-basecalling of existing datasets with higher accuracy models and the resulting implications for downstream analysis.

Second, for real-time sequencing, high basecalling throughput is a critical factor [7]. In particular, scenarios such as *field sequencing* [40] and *adaptive sampling* [49] necessitate rapid basecalling due to hardware limitations and the need for real-time decision-making. Field sequencing, often conducted in remote or resource-constrained environments, demands immediate basecalling to obtain actionable genomic information swiftly. Conventional high-compute infrastructure is often unavailable or impractical in these settings, underscoring the importance of an efficient basecalling process. Similarly, adaptive sampling protocols, aiming to optimize sequencing output based on real-time analysis of initial sequencing data, require a fast and accurate basecaller to make prompt decisions regarding read continuation or rejection. Also, enhancing the speed and efficiency of basecalling is critical for re-basecalling existing datasets using advanced, higher-accuracy models. By revisiting earlier data with improved basecalling algorithms, researchers can achieve a more precise representation of the genomic sequence. Current basecallers provide a tradeoff between

speed and accuracy, often leading to sub-optimal performance in real-time sequencing scenarios.

Second, we have included runtime estimates of basecalling in the complete genome sequencing pipeline
in Section **1 Background**:

We observe that basecalling is the single longest stage in the genome sequencing pipeline, taking up
to 43% of execution time while the subsequent step of overlap finding, assembly, read mapping, and
polishing take 18%, 4%, <1%, and 35% of execution time, respectively.

By incorporating these changes, our revised manuscript effectively communicates the importance of
faster and more efficient basecalling in specific practical scenarios, aligning with your valuable feedback.

*[IR1.2] The authors state: For example, Bonito_CRF-fast, a throughput-optimized recurrent neural
network-based version of Bonito [45] basecaller from ONT, takes ~6 hours to basecall a 3 Gbps (Giga
base pairs) human genome on a powerful server-grade GPU. I am a bit surprised by how slow that is,
since in this published benchmark by ONT ([https://aws.amazon.com/blogs/hpc/benchmarking-
the-oxford-nanopore-technologies-basecallers-on-aws/](https://aws.amazon.com/blogs/hpc/benchmarking-the-oxford-nanopore-technologies-basecallers-on-aws/)), they show that it can take at the
slowest 62 hours to basecall 90 Gbps of data, which would translate that you would need at max 2 hours
(probably less since the benchmark uses the high accuracy model, not the fast one) to basecall 3 Gbps.
This is with a reasonable hardware setup: 1GPU with 16Gb VRAM, 16Gb RAM and 4 CPUs. I would
encourage the authors to revise why there are such discrepancies between their results and the ones from
ONT.
33*

We have addressed this comment as a part of Common Comments 1.4.

*[IR1.3] There is a lack of information on the methods section regarding the processing of the data
before and after basecalling. Raw nanopore signals can be hundreds of thousands long which makes
it not feasible to use as input directly to the neural network. The nanopore raw signal is normalized
(not discussed in the manuscript), chunked (not discussed in the manuscript) into small pieces (usually
overlapping) and then fed into the basecaller. Then, the output of the neural network is stitched
back together (not discussed in the manuscript) and decoded (not discussed in the manuscript). Some
parameters, like chunk size, overlap and decoding method (greedy, beam-search) are completely omitted
in the methods.
47*

To address this comment, we have revised Section **5 Methods** in three ways. First, we provide a detailed
explanation of the preprocessing steps for raw nanopore signals. This includes normalizing and chunking
the signals into manageable, overlapping segments, preparing them for input into the neural network.

Prior to basecalling, raw nanopore signals undergo a preprocessing pipeline to prepare them for input
into the neural network. Raw nanopore signals, which can be hundreds of thousands of data points
long, are normalized to ensure consistent input characteristics for the subsequent processing steps.
We use empirically determined normalization scaling factors from ONT's Bonito_CTC basecaller.
The normalized signals are chunked into smaller segments, typically with overlapping regions.

Second, we have now elaborated on the post-processing steps after basecalling, explicitly discussing the stitching together of neural network output segments and the subsequent decoding process to obtain the final basecalled sequences. We also mention our used decoding method (i.e., beam-search).

After basecalling, the basecalled sequences obtained from individual signal segments are stitched back together to reconstruct the entire nucleotide sequence. The stitched sequences are then decoded to obtain the final basecalled sequences. We use the beam-search decoding [106] method to obtain the final basecalled sequences from stitched segments.

Third, we provide a description of critical parameters such as chunk size and overlap. Their significance and impact on the overall basecalling process are now clearly outlined in the methods section.

The chunk size and overlap are empirically set to 4000 bps and 500, respectively. Chunk size affects the balance between processing speed and accuracy. Smaller chunk sizes can lead to more accurate basecalling but may require more computational resources and time. Larger chunk sizes may be faster but can potentially introduce errors if the signal varies significantly within the chunk. Overlap represents the degree to which consecutive chunks share data with each other. Overlapping chunks can help mitigate the potential issues caused by abrupt changes in the signal at chunk boundaries. It allows for a smoother transition between chunks, reducing the chances of missing important information in the signal. However, a larger overlap may increase computational demands and processing time.

[IR1.4] The introduction is quite verbose. I certainly see that this intro would be necessary for the average GB reader, as many computational biologists will be unfamiliar with many of these concepts. However, I believe some of it should either be shortened or moved to the methods, to speed up the transition to the results. For example, SkipClip requires 4 paragraphs to describe, which in my opinion, is the simplest of the approaches to explain. The introduction also uses 4 figure panels, I would suggest having a single panel that quickly summarizes the different approaches, and provide more detailed figures in the supplementary material.

We appreciate your suggestions for streamlining and improving the clarity of the introductory section. In our revised manuscript, we have made the following three changes. First, we use a single informative figure in Section **1 Background** and have moved the detailed figures and descriptions to the supplementary material for reference. Second, we have streamlined the description of SkipClip in Supplementary Section S2 to provide a more concise yet comprehensive understanding. Third, we updated our manuscript template to the Genome Biology Methods template (as requested by the editor) and have significantly shortened the abstract (100 words) to make it more concise and expedite the transition to the results. We will further refine the text and improve the clarity of the paper for the final camera-ready version.

[IR1.5] Although the main goal of the manuscript is the development of accurate and very fast basecallers; and for this reason the comparison of RUBICALL is to also fast basecallers. Despite this, it would be good to have an idea of what would be the "maximum" achievable performance. This is done on Figure S3, where Bonito SUP is included; however, the performance of Bonito SUP is not mentioned

anywhere in the main text, which is a missed opportunity.

We thank the reviewer for this comment. We updated the main section **1 Background** to mention Bonito-SUP explicitly.

Fourth, we show in Supplementary S4 that compared to the most accurate state-of-the-art basecaller (i.e., *Bonito_CRF-sup*), RUBICALL provides $185.54\times$ speedup using $19.22\times$ lower parameters at the expense of, on average, 2.47% lower accuracy.

[IR1.6] I might have missed something, but it is not entirely clear why for some analysis only RUBICALL is compared against the other basecallers, and not RUBICALL-FP and RUBICALL-MP. For example Fig9 has both, but Fig10 has only RUBICALL. It would be best to have both in all figures. If there's a good reason for that not to happen, then properly annotate if RUBICALL means RUBICALL-MP or RUBICALL-FP.

RUBICALL -FP is simply the floating-point variant of RUBICALL -MP. Since RUBICALL -FP and RUBICALL -MP use the same model architecture and weights, RUBICALL -MP, with its mixed precision architecture, provides the same accuracy as RUBICALL -FP. Therefore, both the basecallers lead to the same results, and we refer to them as RUBICALL. In **2.4 Basecalling Accuracy**, we already explicitly mention the above information to provide clarity and transparency in the representation of the results. In our revised manuscript, we keep RUBICALL -FP and RUBICALL -MP separate in Figures 7 and Figure 8 to maintain consistency, which were Figure 10 and Figure 11 in our initial submission, respectively.

Figure 7: Basecalling accuracy comparison of RUBICALL (using floating-point precision (RUBICALL-FP) and mixed-precision (RUBICALL-MP)).

Figure 8: Comparison of RUBICALL (using floating-point precision (RUBICALL-FP) and mixed-precision (RUBICALL-MP)) for normalized (a) mismatches, (b) bases mapped, (c) reads mapped, and (d) reads unmapped.

[IR1.7] References to Figure 1 and 2 use numeric systems, instead of the traditional letter system. References should read (Fig 1a, Fig1b, etc.)

References to Figures 1 and 2 have been updated to use the traditional letter system. In the revised manuscript, Figure 2 is Figure S1, while Figure 1 is still Figure 1.

[IR1.8] Figure 13 and figure 15 need text explaining the figure.

We updated our explanation of Figure 13 (Figure 9 in the revised manuscript) as follows:

Figure 9 shows the effect of SkipClip on validation accuracy using three different strides at which we remove a skip connection from a block, i.e., the epoch interval at which SkipClip removes a skip connection from a block. We use our QABAS-designed model that has five blocks of skip connections. We highlight the number of epochs needed to remove all the skip connections for different strides. For example, Stride 1 requires five epochs to remove all the skip connections, while Stride 3 requires fifteen epochs.

We updated our explanation of Figure 15 (Figure 11 in the revised manuscript) as follows:

We highlight each layer’s precision (i.e., weights and activation precision) using distinct colors. Our range includes floating-point (i.e., fp32) computation to integer computation (i.e., int16, int8, and

int4) for weight and activation. Based on our experiments in Section 2.1.2, we restrict the precision of weight and activation in RUBICALL-MP architecture in QABAS to int8 and int4, respectively.

[IR1.9] Figure S1 denotes the change in number of parameters in gray inside the plot frame box. I find it a bit confusing since it seems that the baseline model has 94.63 parameters, but that value is the accuracy. I suggest to move the #parameters as well as the values outside the plot frame box.

In Figure S1, which is Figure S3 in the revised manuscript, the number of parameters and values have been moved outside the plot frame box to avoid confusion.

Figure S3: Basecaller sensitivity to skip connections.

[IR1.10] Figure S2 I assume also talks about validation accuracy in Bonito_CTC architecture, might be best to also explicitly mention that.

Text for Figure S2, which is Figure S4 in the revised manuscript, now explicitly mentions that it is related to validation accuracy in the Bonito_CTC architecture.

In Figure S4, we show the effect of two critical hyper-parameters of SkipClip (alpha (α) and temperature (τ)) on the validation accuracy of Bonito_CTC.

[IR1.11] Figure S3c, seems a bit counter-intuitive to have basecalling accuracy axis flipped.

As mentioned in Common Comments 1.3, the basecalling accuracy axis in Figure S3c, which is Figure S6 in the revised manuscript, has been adjusted.

[IR1.12] Table S1, the basecalling accuracy metrics for RUBICALL-FP and RUBICALL-MP are exactly the same, this might be true, but perhaps double-check this is not a copy-paste error.

As discussed in IR1.6, RUBICALL-FP and RUBICALL-MP use the same model architecture and weights. We ensure that RUBICALL-MP, with its mixed precision architecture, provides the same accuracy as RUBICALL-FP. Therefore, both the basecallers lead to the same results. Table S1 has been double-checked to ensure that the basecalling accuracy metrics for RUBICALL-FP and RUBICALL-MP are accurate and not a copy-paste error.

[IR1.13] Figure numbers are incorrect: there is no figure 6 nor 11.

Thank you for noticing this error. Figure 11 was marked as Figure 12, while Figure 6 was already correct.
We corrected the figure numbering to ensure that there are no references to a non-existent Figure 11.

[IR1.14] Table 1 has some values in bold, the meaning of these (I assume it's best performing) is not
explained in the table text.

We explained the meaning of bold values in Table 1, specifying that they indicate the best-performing
values. We updated the text in Section 2.5.1 **De Novo Assembly** as follows:

For Genome Fraction (%), Average Identity (%), and Quality Value (QV), we highlight the
highest achieved value. While for Assembly Length, Average GC (%), and NG50, we highlight
the value closest to the real assembly length. For Total Indels and Indel Ratio (%), the best-
performing basecaller has the lowest value.

[IR1.15] Table 1, for assembly length I assume the value closest to the real assembly length (based
on the reference) is best. However, the bold number does not always correspond to that. For example,
for *Haemophilus haemolyticus* M1C132_1, RUBICALL is bold, but the values of Dorado-fast and
Bonito-CRF-fast are closest to the reference assembly length.

We clarified the criteria for bold values in Table 1 regarding assembly length, ensuring they correspond
to the closest values to the reference assembly length. We have made the value of Bonito-CRF-fast bold
for *Haemophilus haemolyticus* M1C132_1. We have double-checked all the other values.

[IR1.16] Table 1, *Klebsiella pneumonia* INF032 and *Serratia marcescens* 17-147-1671, the assembly
length and NG50 for SACall values are not formatted properly as they lack the thousands comma
separators.

We have properly formatted the assembly length and NG50 values with thousands comma separators
for *Klebsiella pneumonia* INF032 and *Serratia marcescens* 17-147-1671 in Table 1.

[IR1.17] Table 1, for many cases the Assembly length and NG50 values are exactly the same. This could
be correct, but I would encourage the authors to re-check just in case these are not copy-paste mistakes.

We have reviewed the accuracy of Assembly length and NG50 values in Table 1 to prevent copy-paste
mistakes. We confirm that the Assembly length and NG50 values are exactly the same for the bacterial
species.

Reviewer 2

[IR2.1] Generalization: The work was mainly conducted on AMD GPU and XILINX accelerator platforms, which are not commonly used in most labs, where X86+Nvidia GPUs are more commonly used at the current time point. Although the proposed method is not platform-specific, authors are expected to demonstrate the performance of their framework on other different platforms.

Please refer to Common Comments 1.2 for our newly added results on using NVIDIA A40 GPU.

[IR2.2] Pipeline Component: The two major components of the proposed method involve creating a compact network structure for a basecalling model while preserving the model’s efficacy. Could step-2 not be resolved by step-1, considering that the skip-connection could also be treated as a model component, and step-1 aims to find the best model structure? Does this indicate that the QABAS in step-1 can only search for certain specific model architectures? Or to say, are steps 1 and 2 addressing the same problem with different tools (NAS, distillation)?

Thank you for your insightful comment regarding the relationship between the two major components of our proposed method, QABAS (Quantization-Aware Basecalling Neural Architecture Search) and SkipClip.

QABAS and SkipClip indeed aim to address the overarching goal of creating a compact network structure for basecalling without compromising its accuracy. However, they approach this goal from different perspectives and utilize different tools to achieve complementary optimizations. We would like to highlight two critical points because of which we cannot merge the two methods.

First, skip connections play a vital role in preventing vanishing gradients and ensuring proper training of the basecalling model [53]. By maintaining the skip connections in the architecture during the initial NAS phase, we guarantee that the model’s training is stable and effective. It is only after the QABAS and the final training of the selected topology that we introduce SkipClip. At this stage, we apply SkipClip to gradually remove the skip connections in a controlled manner. If we were to combine QABAS and SkipClip into a single step, we would risk not finding the optimal solution achieved by our two-step approach. This separation allows us to benefit from the strengths of both techniques – QABAS in optimizing the architecture for hardware efficiency and SkipClip in gradually removing skip connections – to achieve a more robust and effective basecalling architecture.

Second, QABAS focuses on tailoring the basecalling neural network architecture for a specific hardware acceleration platform while optimizing the bit-width precision for each layer WITHOUT a teacher network. Using NAS techniques, QABAS explores various architectural configurations and identifies the most suitable network architecture for the given hardware. SkipClip, on the other hand, leverages knowledge distillation to train a smaller network (student) without skip connections to mimic a pre-trained larger network (teacher) that utilizes skip connections. The teacher network provides an upper bound on the achievable accuracy. It is essential to note that: (1) QABAS, while capable of incorporating Step 1, may not inherently focus on skip connection removal and lead to an architecture with skip connections, and (2) in QABAS, we do not restrict ourselves to a specific teacher network architecture, hence the final network is not bounded by teacher’s accuracy. SkipClip’s knowledge distillation approach specifically addresses skip connection removal, which may not be efficiently addressed by the architecture-focused QABAS alone.

In summary, both steps aim to achieve a compact and efficient basecalling model, but they do so through different tools and approaches (NAS and knowledge distillation), ensuring a comprehensive optimization process.

To better clarify this distinction and synergy between QABAS and SkipClip, we have added the below discussion point in Section 3 **Discussion**:

Separation between QABAS and SkipClip. Both QABAS and SkipClip share the overarching objective of creating a compact basecalling network without compromising accuracy. However, they approach this goal from distinct perspectives and employ different optimization tools. The following three points justify the separation of the two methods. First, skip connections are integral to stable model training, and by retaining them during the initial QABAS phase, we ensure effective training of the final basecalling network. The subsequent application of SkipClip allows for the controlled removal of skip connections, contributing to a more robust solution. Second, QABAS might find an architecture with skip connections, whereas SkipClip employs knowledge distillation for skip connection removal, addressing a specific aspect not efficiently handled by QABAS alone. Third, unlike SkipClip, QABAS tailors the neural network architecture for hardware efficiency without relying on a teacher network. The teacher network provides an upper bound on the achievable accuracy. Therefore, this two-step approach optimally combines the strengths of NAS and knowledge distillation, ensuring a comprehensive and effective optimization process for a compact and efficient basecalling model.

[IR2.3] Application Scenario: The proposed framework is optimized with hardware constraints in mind, but this optimization incurs a computational cost. Different end users may use different hardware. Although the optimized model can be highly efficient in the decoding stage, users are required to optimize the neural network model according to their specific hardware and then re-train the basecaller. Can the authors provide a more universal hardware setting for direct deployment? Besides being hardware-friendly, user-friendliness is also a crucial consideration.

Thank you for your valuable feedback. We appreciate your understanding of the framework’s flexibility in adapting to different hardware configurations. Indeed, users are encouraged to tailor the hardware setting to their specific devices and requirements to maximize performance. While we optimize our framework for our target hardware, i.e., AIE, we are aware that hardware choices can vary among end users. To accommodate this, we have designed our framework using the open-source nn-Meter (<https://github.com/microsoft/nn-Meter>) tool in a way that allows users to freely configure and adapt the hardware settings according to their particular devices by means of the `applied_hardware` flag in RUBICON. nn-Meter offers a wide range of hardware models for various devices, enabling users to select or build custom models that suit their hardware configurations. This integration enhances the adaptability and user-friendliness of our framework, allowing for efficient optimization and deployment across diverse hardware environments. Second, in our open-source RUBICON code [127], we have comprehensive documentation and tutorial notebooks that guide users through the process of setting up and optimizing the framework for various hardware configurations. This will ensure that users have the necessary resources to implement the basecaller according to their specific needs efficiently.

To address this comment, we have added the following explanation in the Supplementary **S1 Quantization-Aware Basecaller Architecture Search (QABAS)**:

We provide an example latency estimator for our target hardware (i.e., AIE) in RUBICON. However,
our integration with the open-source nn-Meter [98] tool allows users to freely configure hardware
settings through the `applied_hardware` flag in RUBICON. This integration enhances adaptability,
enabling efficient optimization and deployment across different hardware environments.

*[IR2.4] Evaluation of Datasets: The study mainly evaluated nanopore R9.4 reads from nine prokaryotic*
*species. A more comprehensive result might be achieved by using other benchmark datasets (e.g., HG002)*
*and the latest R10.4 chemistry with publicly available datasets.*

Please refer to Common Comments 1.1 for our new results on HG002.

Regarding using R10 chemistry datasets: Currently, ONT does not provide an appropriate public training
dataset for R10 chemistry. They only provide in-house trained R10 models, which cannot be used to have
a consistent evaluation of all the basecallers. Since R9 and R10 chemistry use different generations of
nanopore technologies (e.g., different pore protein and read length), we cannot use R9-trained models for
inference on R10 sequenced datasets. Therefore, we use only the available R9 chemistry training dataset
from ONT and perform inference on R9 chemistry datasets. We had already raised an issue to ONT on
October 25, 2021 (<https://github.com/nanoporetech/bonito/issues/195>).

We have updated our paragraph related to **Basecalling reads** in Section 5 **Methods** as follows:

We use only R9 chemistry datasets as, currently, ONT does not provide a suitable public training
dataset for R10 chemistry. They offer in-house trained R10 models that cannot be employed for a
consistent evaluation across all basecallers. R9 and R10 chemistries involve distinct generations
of nanopore technologies, including different pore proteins and read lengths. Therefore, models
trained on R9 chemistry are incompatible for inference on R10 sequenced datasets. Due to these
technical constraints, our study is currently limited to utilizing the available R9 chemistry training
dataset from ONT and conducting inference exclusively on R9 chemistry datasets.

Reviewer 3

[IR3.1] It might be good to explain the specifics of data being discussed in the introduction. For example, I was not sure if the authors meant basecalling 3 Gb of sequence (1x human genome) takes 6 hours using Bonito.

We have addressed this comment to provide specifics of the data being discussed as a part of Common Comments 1.4.

[IR3.2] It might also be good to define performance on other GPUs which, I think, would be relevant for users. Was the performance measured on NVIDIA/ARM?

Please refer to Common Comments 1.2 for our newly added results on using NVIDIA A40 GPU.

[IR3.3] I would not call bonito as state-of-the-art. It was the R&D basecaller that ONT released to the community. However, the SOTA for Nanopore at the time would have been Guppy (and now a new version of Dorado). I appreciate the source code for those is not open. I do, however, feel that the performance (in runtime, speed, and quality) would benefit from being measured relative to those for users in the community.

Thank you very much for your insightful comment. We already provide runtime measurements for the Dorado basecaller v0.4.0 in all our evaluation results (Section 2 **Results**). In our revised manuscript, we have ensured that Bonito is not referred to as a state-of-the-art basecaller.

[IR3.4] Figure 8c. It is odd to see a decrease in accuracy on the axis (which makes it hard to follow).

Please refer to Common Comments 1.3. We have updated Figure 8c, which is Figure 5(a) in the revised manuscript, with a proper axis.

[IR3.5] It may also be worth breaking down the performance comparisons (possibly as a supplementary figure) to compare fast vs. fast and higher accuracy (e.g. Bonito-CTC, or sup) with high accuracy models.

We thank the reviewer for this feedback. We have updated Table S1 in supplementary Section S4 with an additional column where we compare the main characteristics of different basecallers in terms of accuracy and performance.

Table S1: Comparison of RUBICALL-MP with baseline basecallers in terms of model architecture, characteristics, precision, basecalling throughput, basecalling accuracy, parameters, and model size. For basecalling throughput and basecalling accuracy, we report average (Avg.), minimum (Min.), maximum (Max.), 25th percentile (25th %tile), and 75th percentile (75th %tile) values for all the basecallers.

Basecaller	Architecture	Characteristics	Precision	Basecalling Throughput (kbp/sec)					Basecalling Accuracy (%)					Parameters	Model Size (MB)
				Avg.	Min.	Max.	25th %tile	75th %tile	Avg.	Min.	Max.	25th %tile	75th %tile		
Causalcall	CNN	Low Accuracy	FP32	26.76	11.04	53.94	18.65	30.13	84.02	82.70	86.42	83.57	86.17	3,589,893	13.69
SACall	Transformer	Low Accuracy	FP32	119.71	47.33	346.30	86.32	112.58	87.28	86.44	91.44	86.97	89.22	9,854,725	37.59
Bonito_CTC	CNN	High Accuracy	FP32	76.22	31.68	219.42	54.64	73.46	89.19	87.99	93.75	88.62	91.15	9,738,573	37.15
RUBICALL-FP	CNN	High Accuracy	FP16	194.74	99.8	394.9	129.72	240.34	89.25	86.59	93.64	88.97	91.41	3,314,578	12.64
Bonito_CRF-sup	RNN	Highest Accuracy	FP16	52.63	20.22	149.2	35.03	55.56	91.73	90.60	95.95	91.43	93.72	26,992,744	103.03
Bonito_CRF-fast	RNN	Fast Performance	FP16	685.13	261.53	1044.37	421.52	881.84	86.36	82.53	91.39	86.25	88.51	730,344	2.79
Dorado-fast	RNN	Fast Performance	FP16	2593.34	1155.06	3927.03	1835.99	3351.95	87.16	82.53	91.39	86.25	88.51	730,344	2.79
RUBICALL-MP	CNN	High Accuracy and Fast Performance	Mixed-Precision	9765.65	5309.74	12862.73	8433.53	10892.55	89.25	86.59	93.64	88.97	91.41	3,314,578	5.36

Based on this new column in Table S1, we added the following text in Section S4:

RUBICALL-MP is the only basecaller that provides both higher performance and accuracy when compared to all the other evaluated basecallers.

[IR3.6] It would be useful to compare under and over represented sequences (kmers) for the reads and resulting assemblies.

We thank the reviewer for this feedback. We added a new Supplementary Section **S7 K-mer Counting Analysis** where we perform k-mer frequency analysis for reads and assemblies.

S7. K-mer Counting Analysis

We analyze the occurrence of k-mer (i.e., substrings of length k) in a given sequence of basecalled reads and their assemblies in Figure S8 and Figure S9, respectively. We use BMap [130] to collect the number of unique k-mers and the frequency of each unique k-mer in a given sequence. During our analysis, we vary the value of k from 15 to 31. Based on our empirical analysis, we set the k value for our evaluated bacterial species to 15, where we observe distinct peaks of unique k-mers. We do not perform k-mer frequency analysis for the human genome due to the low coverage of the human genome in our experiments. We make the following two observations from Figure S8 and Figure S9. First, RUBICALL has distinct peaks for all the evaluated species, often matching the k-mer composition generated from Bonito_CTC. Second, Bonito_CRF-fast and Dorado-fast generate similar k-mer compositions as they both have the same neural network architecture.

Figure S8: K-mer frequency analysis of generated reads from RUBICALL and all the other evaluated basecallers.

Figure S9: K-mer frequency analysis of generated assemblies of reads from RUBICALL and all the other evaluated basecallers.

Table S3 presents an analysis of k-mer frequencies in the raw reads and the corresponding assemblies. We include common sequences and read-to-assembly ratios to provide a comprehensive view of the similarities and disparities in sequence representation, aiding in assessing data quality and the performance of the assembly algorithms. We observe that the k-mers identified as over-represented in the assemblies are mainly observed as over-represented k-mers in read sets for most basecallers. These over-represented k-mers are likely to appear due to the particular repetitive regions of each genome, making k-mers appear a larger amount of times for these regions. Therefore, there is potentially no additional insertion or depletion of these k-mers during the assembly process.

Table S3: Comparison of under and over-represented sequences (k-mers) in reads and assemblies for all the evaluated basecallers. For both under and over-represented sequences, we show common sequences (Common) and the ratio of k-mer frequencies between reads and assemblies (Ratio).

Dataset	Basecaller	Under-Represented				Over-Represented			
		Read	Assembly	Common	Ratio	Read	Assembly	Common	Ratio
Acinetobacter pittii 16-377-0801	Causalcall	73,096,681	3,768,504	3,221,668	0.855	6,263	187	187	1.000
	Bonito_CRF-fast	55,359,526	3,747,817	2,367,806	0.632	17,983	360	360	1.000
	Bonito_CTC	44,790,782	3,593,419	1,814,762	0.505	29,097	296	296	1.000
	SACall	55,660,535	3,625,236	2,381,232	0.657	15,534	368	368	1.000
	Dorado-fast	55,775,603	3,760,029	2,404,108	0.639	18,137	425	425	1.000
RUBICALL	44,085,891	3,609,296	1,793,772	0.497	30,316	430	430	1.000	
Haemophilus haemolyticus M1C132_1	Causalcall	31,021,572	NA	NA	NA	1,552	NA	NA	NA
	Bonito_CRF-fast	42,355,232	2,077,823	2,076,203	0.999	2,865	54	53	0.981
	Bonito_CTC	35,847,257	1,919,667	700,997	0.365	33,713	94	94	1.000
	SACall	36,998,888	2,000,357	1,998,679	0.999	22,517	98	61	0.622
	Dorado-fast	41,939,332	2,074,317	2,072,740	0.999	4,714	37	37	1.000
RUBICALL	33,917,316	1,929,302	1,127,668	0.584	23,221	100	100	1.000	
Klebsiella pneumoniae INF032	Causalcall	197,327,230	4,850,481	3,833,326	0.790	15,891	6	6	1.000
	Bonito_CRF-fast	169,124,267	4,914,464	3,353,593	0.682	35,175	22	22	1.000
	Bonito_CTC	155,835,445	4,758,242	2,718,894	0.571	53,149	20	20	1.000
	SACall	176,768,581	4,747,776	3,366,011	0.709	28,038	12	12	1.000
	Dorado-fast	167,899,392	4,916,810	3,351,520	0.682	35,059	24	24	1.000
RUBICALL	150,348,189	4,776,357	2,689,193	0.563	62,356	26	26	1.000	
Klebsiella pneumoniae INF042	Causalcall	211,565,073	5,171,081	4,222,032	0.816	22,204	3	3	1.000
	Bonito_CRF-fast	178,074,237	5,212,872	3,532,743	0.678	61,454	29	29	1.000
	Bonito_CTC	162,568,221	4,964,190	2,843,401	0.573	81,948	36	36	1.000
	SACall	186,186,165	5,024,228	3,609,853	0.718	40,440	41	41	1.000
	Dorado-fast	174,755,139	5,508,965	3,808,628	0.691	63,644	23	23	1.000
RUBICALL	158,433,298	4,989,328	2,818,523	0.565	92,045	37	37	1.000	
Klebsiella pneumoniae KSB2_1B	Causalcall	180,267,220	5,064,568	4,648,903	0.918	8,844	6	6	1.000
	Bonito_CRF-fast	152,878,553	5,122,808	4,044,261	0.789	23,755	16	16	1.000
	Bonito_CTC	137,461,268	4,859,560	3,174,990	0.653	27,211	14	14	1.000
	SACall	158,736,471	4,903,831	4,117,531	0.840	16,090	12	12	1.000
	Dorado-fast	150,458,414	5,265,881	4,164,287	0.791	24,938	19	19	1.000
RUBICALL	134,066,854	4,876,789	3,186,025	0.653	31,451	17	17	1.000	
Klebsiella pneumoniae NUH29	Causalcall	140,405,375	5,060,601	5,004,375	0.989	835	1	1	1.000
	Bonito_CRF-fast	110,315,181	5,060,829	4,696,878	0.928	4,320	22	22	1.000
	Bonito_CTC	97,405,757	4,775,503	4,182,794	0.876	5,177	20	20	1.000
	SACall	112,996,097	4,844,709	4,613,301	0.952	2,941	16	16	1.000
	Dorado-fast	108,585,877	5,043,253	4,679,036	0.928	4,483	23	23	1.000
RUBICALL	95,201,166	4,789,453	4,136,482	0.864	5,645	25	25	1.000	
Serratia marcescens 17-147-1671	Causalcall	66,514,376	5,334,807	5,193,034	0.973	30,238	4	4	1.000
	Bonito_CRF-fast	63,963,265	5,399,858	4,554,600	0.843	61,321	4	4	1.000
	Bonito_CTC	53,413,056	5,217,101	3,821,412	0.732	61,275	9	9	1.000
	SACall	64,337,585	5,284,226	4,534,343	0.858	54,872	1	1	1.000
	Dorado-fast	63,535,166	5,451,215	4,583,027	0.841	62,006	4	4	1.000
RUBICALL	51,724,568	5,243,385	3,741,711	0.714	64,943	9	9	1.000	
Staphylococcus aureus CAS38_02	Causalcall	106,477,765	2,791,690	2,784,563	0.997	3,375	3	3	1.000
	Bonito_CRF-fast	72,908,007	2,813,307	2,774,962	0.986	12,170	33	33	1.000
	Bonito_CTC	59,047,673	2,736,120	2,669,644	0.976	18,475	37	37	1.000
	SACall	73,372,106	2,741,321	2,710,663	0.989	10,487	29	29	1.000
	Dorado-fast	73,708,021	2,824,623	2,786,854	0.987	11,902	37	37	1.000
RUBICALL	58,939,315	2,739,823	2,675,209	0.976	18,186	84	84	1.000	
Stenotrophomonas maltophilia 17_G_0092_Kos	Causalcall	183,625,102	4,653,477	4,000,323	0.860	25,103	132	132	1.000
	Bonito_CRF-fast	144,980,026	4,647,752	3,083,896	0.664	127,258	210	210	1.000
	Bonito_CTC	127,334,549	4,383,078	2,495,090	0.569	170,398	285	285	1.000
	SACall	139,640,543	4,422,404	3,045,239	0.689	112,124	201	201	1.000
	Dorado-fast	143,087,662	4,594,794	3,087,855	0.672	127,653	201	201	1.000
RUBICALL	121,924,168	4,391,308	2,430,608	0.554	197,883	327	327	1.000	

We added a reference to this above analysis in Section 2 **Results** as follows:

We also collect the number of unique k-mers and the frequency of each unique k-mer in a given sequence to perform a comparison of under and over-represented k-mers in Supplementary Section S7.

[IR3.7] The QVs should be compared as well to ascertain that RUBICALL data based assemblies are not only covering higher percent of the genome but also better in consensus quality.

We acknowledge the importance of measuring the reliability of the basecalls. In our revised manuscript, we added new experiments to include the Quality Value (QV) score of the assembly in Table 1 in Section 2.5.1 **De Novo Assembly**.

Table 1: Assembly quality comparison of the evaluated basecallers for different species. We measure assembly accuracy in terms of genome fraction (Genome Fraction (%)) and average identity (Average Identity (%)). Genome fraction is the portion of the Reference genome that can align to a given assembly, while average identity is the average of the identity of assemblies when compared to their respective Reference genomes. We measure statistics related to the contiguity and completeness of the assemblies in terms of the overall assembly length (Assembly Length), Average GC content (Average GC (%)) (i.e., the ratio of G and C bases in an assembly), NG50 statistics (NG50) (i.e., shortest contig at the half of the overall Reference genome length), total number of indels in all aligned bases in the assembly (Total Indels), the ratio of indels to assembly length (Indel Ratio (%)), and the reliability of basepairs using the quality value (Quality Value). NA indicates that the generated assemblies were unalignable to the reference genome.

Dataset	Basecaller	Genome Fraction (%)	Average Identity (%)	Assembly Length	Average GC (%)	NG50	Total Indels	Indel Ratio (%)	Quality Value (QV)
Acinetobacter pitti 16-377-0801	Causalcall	92.45	86.18	3,826,077	42.23	3,826,077	270,228	7.06	11.99
	Bonito_CRF-fast	96.64	89.29	3,628,317	38.82	3,628,317	242,373	6.68	12.03
	Bonito_CTC	96.87	91.44	3,676,821	38.9	3,676,821	210,496	5.72	12.45
	SACall	96.68	89.42	3,699,232	38.7	3,699,232	247,997	6.7	12.1
	Dorado-fast	96.37	88.72	3,839,847	39.09	3,839,847	245,016	6.38	12.03
	RUBICALL	96.87	91.51	3,694,086	38.82	3,694,086	208,748	5.65	15.42
Reference		100	100	3,814,719	38.78	3,814,719	0	0	-
Haemophilus haemolyticus M1C132_1	Causalcall	0.00	0.00	0	0	0	0	0	NA
	Bonito_CRF-fast	88.76	91.51	2,046,024	37.98	2,046,024	128,481	6.28	12.25
	Bonito_CTC	96.87	90.70	1,957,480	38.87	1,957,480	118,253	6.04	15.34
	SACall	90.11	88.45	2,032,994	38.22	1,880,730	134,702	6.63	13.15
	Dorado-fast	89.42	88.97	2,110,860	39.49	2,110,860	129,503	6.14	12.38
	RUBICALL	96.87	90.54	1,966,781	38.92	1,966,781	119,777	6.09	15.37
Reference		100	100	2,042,591	38.46	2,042,591	0	0	-
Klebsiella pneumoniae INF032	Causalcall	92.45	87.35	4,959,127	56.9	4,959,127	353,550	7.13	10.54
	Bonito_CRF-fast	92.69	87.53	4,761,297	57.19	4,761,297	347,299	7.29	10.56
	Bonito_CTC	94.50	90.20	4,897,352	56.65	4,897,352	317,428	6.48	11.26
	SACall	93.97	88.08	4,874,880	56.87	4,874,880	379,028	7.78	10.8
	Dorado-fast	93.00	87.69	5,063,562	56.8	5,063,562	348,572	6.88	10.64
	RUBICALL	94.51	90.30	4,924,240	56.85	4,924,240	314,651	6.39	11.27
Reference		100	100	5,111,537	57.63	5,111,537	0	0	-
Klebsiella pneumoniae INF042	Causalcall	91.44	87.36	5,288,166	56.94	5,288,166	374,162	7.08	10.84
	Bonito_CRF-fast	92.08	88.49	5,052,889	56.8	5,052,889	357,354	7.07	10.93
	Bonito_CTC	93.12	90.49	5,111,083	56.61	5,111,083	317,075	6.2	11.40
	SACall	92.93	88.60	5,149,039	56.72	5,149,039	369,388	7.17	11.08
	Dorado-fast	90.21	88.20	5,737,059	56.44	5,401,717	342,141	5.96	10.98
	RUBICALL	93.12	90.60	5,146,050	56.72	5,146,050	312,448	6.07	11.42
Reference		100	100	5,337,491	57.41	5,337,491	0	0	-
Klebsiella pneumoniae KSB2_1B	Causalcall	91.58	86.97	5,175,311	57.09	5,175,311	363,807	7.03	10.88
	Bonito_CRF-fast	90.24	88.00	4,932,626	56.71	4,932,626	357,769	7.25	10.86
	Bonito_CTC	93.07	90.11	5,003,377	56.69	5,003,377	320,519	6.41	11.41
	SACall	93.58	88.19	5,034,408	56.79	5,034,408	372,380	7.4	11.16
	Dorado-fast	90.28	87.67	5,442,186	56.72	5,261,731	349,387	6.42	11.03
	RUBICALL	93.07	89.89	5,023,639	56.75	4,932,626	357,769	7.12	11.25
Reference		100	100	5,228,889	57.59	5,228,889	0	0	-
Klebsiella pneumoniae NUH29	Causalcall	89.08	86.01	5,158,874	56.78	5,158,874	389,676	7.55	11.75
	Bonito_CRF-fast	92.17	89.34	4,942,833	57.01	4,942,833	355,690	7.2	11.47
	Bonito_CTC	94.36	90.26	4,918,147	57.04	4,918,147	324,406	6.6	11.92
	SACall	93.66	88.58	4,978,307	57.06	4,978,307	360,950	7.25	11.56
	Dorado-fast	92.27	88.12	5,195,594	57.01	5,195,594	355,728	6.85	11.56
	RUBICALL	94.36	90.43	4,940,813	57.18	4,940,813	316,019	6.4	11.83
Reference		100	100	5,134,281	57.61	5,134,281	0	0	-
Serratia marcescens 17-147-1671	Causalcall	89.91	86.23	5,532,953	57.86	5,422,052	401,545	7.26	13.39
	Bonito_CRF-fast	96.06	89.56	5,479,812	58.85	5,282,474	345,351	6.3	12.66
	Bonito_CTC	96.76	91.38	5,534,329	58.41	5,316,651	298,982	5.4	13
	SACall	94.29	89.36	5,366,913	58.57	5,366,913	358,954	6.69	12.27
	Dorado-fast	96.51	88.87	5,758,989	58.29	5,282,474	348,968	6.06	12.5
	RUBICALL	96.76	91.59	5,597,251	58.52	5,346,640	294,643	5.26	13.01
Reference		100	100	5,517,578	59.13	5,517,578	0	0	-
Staphylococcus aureus CAS38_02	Causalcall	94.35	87.29	2,849,123	36.59	2,810,038	191,730	6.73	10.8
	Bonito_CRF-fast	96.27	91.49	2,790,895	33.05	2,752,169	149,623	5.36	11.59
	Bonito_CTC	97.03	93.57	2,858,986	32.86	2,819,356	123,542	4.32	12.82
	SACall	95.66	91.25	2,837,503	32.91	2,798,079	165,200	5.82	11.57
	Dorado-fast	96.70	91.16	2,927,882	33.52	2,752,169	152,216	5.2	11.64
	RUBICALL	97.03	93.36	2,860,885	33.24	2,821,276	124,795	4.36	12.59
Reference		100	100	2,902,076	32.82	2,902,076	0	0	-
Stenotrophomonas maltophilia 17_G_0092_Kos	Causalcall	94.85	85.73	4,823,177	63.66	4,823,177	366,228	7.59	11.01
	Bonito_CRF-fast	94.60	89.74	4,596,898	65.5	4,596,898	337,040	7.33	11.10
	Bonito_CTC	95.42	90.14	4,664,226	64.82	4,664,226	298,711	6.4	11.51
	SACall	95.28	88.50	4,672,540	64.98	4,672,540	339,853	7.27	11.11
	Dorado-fast	92.99	87.70	4,854,007	63.99	4,854,007	337,105	6.94	11.01
	RUBICALL	95.46	90.49	4,693,744	65.03	4,693,744	289,073	6.16	11.63
Reference		100	100	4,802,733	66.28	4,802,733	0	0	-
Human HG002	Causalcall	NA	NA	130,962	42.95	13,522	NA	NA	NA
	Bonito_CRF-fast	0.002	92.36	119,570,537	40.34	368,848	2,860	0	18.87
	Bonito_CTC	0.430	95.06	134,732,516	40.86	371,590	384,243	0.29	18.58
	SACall	NA	NA	63,025,520	39.87	320,873	NA	NA	NA
	Dorado-fast	0.001	93.15	121,146,376	39.8	361,677	926	0	17.46
	RUBICALL	0.125	94.50	140,928,248	40.99	393,950	100,256	0.1	17.81
Reference		100	100	2,947,743,500	40.79	2,947,743,500	0	0	-

We use Inspector [115] to calculate the QV score using the generated assembly by a basecaller for a specific read. We updated Section 5 Methods to provide details on our methodology:

We use Inspector [115] to calculate the overall quality value (QV) of an assembly. The QV score
is determined by considering structural and small-scale errors in proportion to the total number
of base pairs in the assemblies. High-quality sequences have higher QV scores, indicating a low
probability of sequencing errors, while low-quality sequences have lower QV scores, suggesting a
higher likelihood of errors.

The high QV score of RUBICALL confirms that RUBICALL assemblies provide reliability by covering a
higher percentage of the genome and better consensus quality. Based on these experiments with QV score,
we added the following observation in Section 2.5.1 De Novo Assembly:

Sixth, RUBICALL consistently provides a higher quality value (QV), indicating a low probability of
sequencing errors. Therefore, compared to the other evaluated basecallers, RUBICALL has higher
reliability of the assembled genome.

*[IR3.8] It would be useful to apply and measure the performance and quality for human genome (or a
chromosome perhaps?).*

Please refer to Common Comments 1.1 for our new results on Human HG002.

*[IR3.9] If the # mapped reads and # bases mapped are comparable across the basecallers, does it suggest
that other basecallers include can call unalignable sequence either within a read or (more likely) as
additional reads?*

We added additional analysis on mapped reads and mapped bases in our new Supplementary Section S6.

S6. Analysis of Mapped Reads and Mapped Bases

Table S2 shows the average read length, the overall number of mapped reads, the number of mapped
bases, and the ratio of mapped bases to the mapped reads. Our goal is to evaluate the tools in terms
of the read lengths they can generate and the alignable fraction of these reads to their corresponding
reference genomes. We make three key observations. First, we find that the average read lengths
are similar across different basecallers for each dataset, except Causalcall for the human genome.
This indicates that the substantial differences in read length are unlikely to influence the ratio of
mapped bases to the number of mapped reads, while the number of alignable sequences within each
read and the number of mapped reads can have the main effect on such a ratio. Second, we find
that basecallers provide a similar number of mapped reads and the ratio of mapped bases to the
mapped reads for each dataset, except Causalcall for the human genome. These similarities mainly
indicate that the unalignable reads and the unalignable regions within each read are likely to be
similar across basecallers, leading to similar ratios of mapped bases to mapped reads when mapping
reads with similar average read lengths. Third, we find that Causalcall provides exceptions for
the human genome in terms of the average read length and the mapped bases to the mapped reads
ratio. This is mainly because Causalcall fails to basecall all raw signals for the human genome and
provides a subset of basecalled reads that other basecallers generate, leading to inaccurate analysis

overall. We conclude that almost all basecallers, except Causalcall, generate reads with similar average read lengths and reads with similar alignable regions, although these similarities differ by certain percentages' as we discuss in Section 2.5.2.

Table S2: Read mapping comparison of RUBICALL with baseline basecallers in terms of mean length of individual sequencing reads in a dataset (Avg. Length), the total number of mapped reads (Mapped Reads), the total number of mapped bases (Mapped Bases), and the ratio of total number of mapped reads and the number of mapped bases.

Dataset	Basecaller	Avg. Length	Mapped Reads	Mapped Bases	#Mapped Bases/ #Mapped Reads
Acinetobacter pittii 16-377-0801	causalcall	25,718.6	4,434	114,159,528	25,746.4
	Bonito_CRF-fast	26,151.3	4,452	110,907,740	24,911.9
	Bonito_CTC	24,879.1	4,457	110,183,466	24,721.4
	SACall	25,153.3	4,451	111,997,940	25,162.4
	Dorado-fast	26,151.1	4,452	110,907,740	24,911.9
	RUBICALL	25,000.8	4,452	111,405,897	25,023.8
Haemophilus haemolyticus M1C132_1	causalcall	NA	NA	NA	NA
	Bonito_CRF-fast	9,835.7	6,444	64,816,196	10,058.4
	Bonito_CTC	8,862.8	6,201	73,573,092	11,864.7
	SACall	6,871.1	4,028	43,233,160	10,733.2
	Dorado-fast	9,844.8	6,444	64,816,196	10,058.4
	RUBICALL	7,751.5	6,287	63,415,299	10,086.7
Klebsiella pneumoniae INF032	causalcall	35,781.9	15,150	542,123,428	35,783.7
	Bonito_CRF-fast	36,556.4	15,147	533,045,454	35,191.5
	Bonito_CTC	35,189.1	15,152	519,659,064	34,296.4
	SACall	35,078.5	15,150	531,456,488	35,079.6
	Dorado-fast	36,624.7	15,147	533,045,454	35,191.5
	RUBICALL	35,420.7	15,153	536,724,002	35,420.3
Klebsiella pneumoniae INF042	causalcall	48,483.8	11,236	542,123,428	48,248.8
	Bonito_CRF-fast	49,617.5	11,252	533,045,454	47,373.4
	Bonito_CTC	46,198.4	11,273	519,659,064	46,097.7
	SACall	46,298.3	11,198	531,456,488	47,459.9
	Dorado-fast	49,621.4	11,252	533,045,454	47,373.4
	RUBICALL	46,637.6	11,268	536,724,002	47,632.6
Klebsiella pneumoniae KSB2_1B	causalcall	24,039.6	16,642	401,041,491	24,098.2
	Bonito_CRF-fast	24,723.9	16,744	384,436,100	22,959.6
	Bonito_CTC	22,918.8	16,803	385,157,295	22,921.9
	SACall	22,917.5	16,371	381,266,978	23,289.2
	Dorado-fast	24,728.0	16,744	384,436,100	22,959.6
	RUBICALL	23,141.9	16,783	388,897,351	23,172.1
Klebsiella pneumoniae NUH29	causalcall	16,233.7	14,954	243,112,795	16,257.4
	Bonito_CRF-fast	16,435.7	15,056	229,123,038	15,218.1
	Bonito_CTC	15,182.0	15,152	233,135,041	15,386.4
	SACall	15,536.5	15,088	234,764,649	15,559.7
	Dorado-fast	16,419.0	15,056	229,123,038	15,218.1
	RUBICALL	15,300.8	15,113	231,523,267	15,319.5
Serratia marcescens 17-147-1671	causalcall	8,198.0	12,729	104,864,058	8,238.2
	Bonito_CRF-fast	8,456.2	16,667	133,916,776	8,034.8
	Bonito_CTC	8,024.8	16,715	133,754,055	8,002.0
	SACall	8,167.1	16,665	136,289,479	8,178.2
	Dorado-fast	8,465.1	16,667	133,916,776	8,034.8
	RUBICALL	8,076.5	16,696	134,916,360	8,080.8
Staphylococcus aureus CAS38_02	causalcall	21,425.2	11,038	236,529,129	21,428.6
	Bonito_CRF-fast	21,932.6	11,047	237,020,597	21,455.7
	Bonito_CTC	21,455.7	11,047	232,091,657	21,009.5
	SACall	21,372.6	11,047	236,103,648	21,372.6
	Dorado-fast	21,930.8	11,047	237,020,597	21,455.7
	RUBICALL	21,501.0	11,047	237,521,476	21,501.0
Stenotrophomonas maltophilia 17_G_0092_Kos	causalcall	31,415.3	15,946	501,018,868	31,419.7
	Bonito_CRF-fast	31,736.6	15,959	470,408,299	29,476.1
	Bonito_CTC	29,453.8	15,997	474,913,094	29,687.6
	SACall	30,095.7	15,985	481,168,698	30,101.3
	Dorado-fast	31,727.6	15,959	470,408,299	29,476.1
	RUBICALL	29,676.7	15,980	474,401,853	29,687.2
Human HG002	causalcall	11,201.7	163,984	2,612,902,733	15,933.9
	Bonito_CRF-fast	37,755.9	238,205	10,627,000,000	44,612.8
	Bonito_CTC	35,831.5	243,686	10,212,000,000	41,906.4
	SACall	32,815.7	203,228	9,080,197,989	44,679.9
	Dorado-fast	37,991.9	238,474	10,627,000,000	44,562.5
	RUBICALL	37,141.2	245,373	10,591,000,000	43,162.9

We added a reference to the above new Supplementary section in Section **2.5.2 Read Mapping**:

We also show the average read length, the overall number of mapped reads and the mapped bases, and the ratio of the number of mapped bases to the number of mapped reads in Supplementary Table S2.

[IR3.10] How do indel errors compare between these basecallers?

We have added two new metrics to our Table 1: (a) Total Indels: the total number of indels in all aligned bases in the assembly, and (b) Indel Ratio (%): the ratio of indels to assembly length. We updated Section **5 Methods** to provide details on these two metrics as additional metrics to measure the assembly accuracy:

3) insertions and deletions of nucleotides (or bases) in the sequence when compared to a reference or other sequences. (i.e., Total Indels and Indel Ratio (%)). Total Indels represents the sum of all the insertions and deletions in the assembled sequence when compared to a reference or other sequences. The Indel Ratio is a measure of the relative abundance of indels compared to the total length of the assembled sequence (calculated using $\text{Total Indels} / \text{Assembly Length} \times 100$). This metric helps to understand the proportion of the assembly that contains insertions and deletions.

We updated Section **2.5.1 De Novo Assembly** as follows. First, we mention that the best-performing basecallers have the lower values:

For Total Indels and Indel Ratio (%), the best-performing basecaller has the lowest value.

Second, we updated Table 1 to include Total Indels and Indel Ratio:

Table 1: Assembly quality comparison of the evaluated basecallers for different species. We measure assembly accuracy in terms of genome fraction (Genome Fraction (%)) and average identity (Average Identity (%)). Genome fraction is the portion of the Reference genome that can align to a given assembly, while average identity is the average of the identity of assemblies when compared to their respective Reference genomes. We measure statistics related to the contiguity and completeness of the assemblies in terms of the overall assembly length (Assembly Length), Average GC content (Average GC (%)) (i.e., the ratio of G and C bases in an assembly), NG50 statistics (NG50) (i.e., shortest contig at the half of the overall Reference genome length), **total number of indels in all aligned bases in the assembly (Total Indels), the ratio of indels to assembly length (Indel Ratio (%)), and the reliability of basepairs using the quality value (Quality Value). NA indicates that the generated assemblies were unalignable to the reference genome.**

Dataset	Basecaller	Genome Fraction (%)	Average Identity (%)	Assembly Length	Average GC (%)	NG50	Total Indels	Indel Ratio (%)	Quality Value (QV)
Acinetobacter pitti 16-377-0801	Causalcall	92.45	86.18	3,826,077	42.23	3,826,077	270,228	7.06	11.99
	Bonito_CRF-fast	96.64	89.29	3,628,317	38.82	3,628,317	242,373	6.68	12.03
	Bonito_CTC	96.87	91.44	3,676,821	38.9	3,676,821	210,496	5.72	12.45
	SACall	96.68	89.42	3,699,232	38.7	3,699,232	247,997	6.7	12.1
	Dorado-fast	96.37	88.72	3,839,847	39.09	3,839,847	245,016	6.38	12.03
	RUBICALL	96.87	91.51	3,694,086	38.82	3,694,086	208,748	5.65	15.42
Haemophilus haemolyticus M1C132_1	Reference	100	100	3,814,719	38.78	3,814,719	0	0	-
	Causalcall	0.00	0.00	0	0	0	0	0	NA
	Bonito_CRF-fast	88.76	91.51	2,046,024	37.98	2,046,024	128,481	6.28	12.25
	Bonito_CTC	96.87	90.70	1,957,480	38.87	1,957,480	118,253	6.04	15.34
	SACall	90.11	88.45	2,032,994	38.22	1,880,730	134,702	6.63	13.15
	Dorado-fast	89.42	88.97	2,110,860	39.49	2,110,860	129,503	6.14	12.38
Klebsiella pneumoniae INF032	RUBICALL	96.87	90.54	1,966,781	38.92	1,966,781	119,777	6.09	15.37
	Reference	100	100	2,042,591	38.46	2,042,591	0	0	-
	Causalcall	92.45	87.35	4,959,127	56.9	4,959,127	353,550	7.13	10.54
	Bonito_CRF-fast	92.69	87.53	4,761,297	57.19	4,761,297	347,299	7.29	10.56
	Bonito_CTC	94.50	90.20	4,897,352	56.65	4,897,352	317,428	6.48	11.26
	SACall	93.97	88.08	4,874,880	56.87	4,874,880	379,028	7.78	10.8
Klebsiella pneumoniae INF042	Dorado-fast	93.00	87.69	5,063,562	56.8	5,063,562	348,572	6.88	10.64
	RUBICALL	94.51	90.30	4,924,240	56.85	4,924,240	314,651	6.39	11.27
	Reference	100	100	5,111,537	57.63	5,111,537	0	0	-
	Causalcall	91.44	87.36	5,288,166	56.94	5,288,166	374,162	7.08	10.84
	Bonito_CRF-fast	92.08	88.49	5,052,889	56.8	5,052,889	357,354	7.07	10.93
	Bonito_CTC	93.12	90.49	5,111,083	56.61	5,111,083	317,075	6.2	11.40
Klebsiella pneumoniae KSB2_1B	SACall	92.93	88.60	5,149,039	56.72	5,149,039	369,388	7.17	11.08
	Dorado-fast	90.21	88.20	5,737,059	56.44	5,401,717	342,141	5.96	10.98
	RUBICALL	93.12	90.60	5,146,050	56.72	5,146,050	312,448	6.07	11.42
	Reference	100	100	5,337,491	57.41	5,337,491	0	0	-
	Causalcall	91.58	86.97	5,175,311	57.09	5,175,311	363,807	7.03	10.88
	Bonito_CRF-fast	90.24	88.00	4,932,626	56.71	4,932,626	357,769	7.25	10.86
Klebsiella pneumoniae NUH29	Bonito_CTC	93.07	90.11	5,003,377	56.69	5,003,377	320,519	6.41	11.41
	SACall	93.58	88.19	5,034,408	56.79	5,034,408	372,380	7.4	11.16
	Dorado-fast	90.28	87.67	5,442,186	56.72	5,261,731	349,387	6.42	11.03
	RUBICALL	93.07	89.89	5,023,639	56.75	4,932,626	357,769	7.12	11.25
	Reference	100	100	5,228,889	57.59	5,228,889	0	0	-
	Causalcall	89.08	86.01	5,158,874	56.78	5,158,874	389,676	7.55	11.75
Serratia marcescens 17-147-1671	Bonito_CRF-fast	92.17	89.34	4,942,833	57.01	4,942,833	355,690	7.2	11.47
	Bonito_CTC	94.36	90.26	4,918,147	57.04	4,918,147	324,406	6.6	11.92
	SACall	93.66	88.58	4,978,307	57.06	4,978,307	360,950	7.25	11.56
	Dorado-fast	92.27	88.12	5,195,594	57.01	5,195,594	355,728	6.85	11.56
	RUBICALL	94.36	90.43	4,940,813	57.18	4,940,813	316,019	6.4	11.83
	Reference	100	100	5,134,281	57.61	5,134,281	0	0	-
Staphylococcus aureus CAS38_02	Causalcall	89.91	86.23	5,532,953	57.86	5,422,052	401,545	7.26	13.39
	Bonito_CRF-fast	96.06	89.56	5,479,812	58.85	5,282,474	345,351	6.3	12.66
	Bonito_CTC	96.76	91.38	5,534,329	58.41	5,316,651	298,982	5.4	13
	SACall	94.29	89.36	5,366,913	58.57	5,366,913	358,954	6.69	12.27
	Dorado-fast	96.51	88.87	5,758,989	58.29	5,282,474	348,968	6.06	12.5
	RUBICALL	96.76	91.59	5,597,251	58.52	5,346,640	294,643	5.26	13.01
Stenotrophomonas maltophilia 17_G_0092_Kos	Reference	100	100	5,517,578	59.13	5,517,578	0	0	-
	Causalcall	94.35	87.29	2,849,123	36.59	2,810,038	191,730	6.73	10.8
	Bonito_CRF-fast	96.27	91.49	2,790,895	33.05	2,752,169	149,623	5.36	11.59
	Bonito_CTC	97.03	93.57	2,858,986	32.86	2,819,356	123,542	4.32	12.82
	SACall	95.66	91.25	2,837,503	32.91	2,798,079	165,200	5.82	11.57
	Dorado-fast	96.70	91.16	2,927,882	33.52	2,752,169	152,216	5.2	11.64
Stenotrophomonas maltophilia 17_G_0092_Kos	RUBICALL	97.03	93.36	2,860,885	33.24	2,821,276	124,795	4.36	12.59
	Reference	100	100	2,902,076	32.82	2,902,076	0	0	-
	Causalcall	94.85	85.73	4,823,177	63.66	4,823,177	366,228	7.59	11.01
	Bonito_CRF-fast	94.60	89.74	4,596,898	65.5	4,596,898	337,040	7.33	11.10
	Bonito_CTC	95.42	90.14	4,664,226	64.82	4,664,226	298,711	6.4	11.51
	SACall	95.28	88.50	4,672,540	64.98	4,672,540	339,853	7.27	11.11
Human HG002	Dorado-fast	92.99	87.70	4,854,007	63.99	4,854,007	337,105	6.94	11.01
	RUBICALL	95.46	90.49	4,693,744	65.03	4,693,744	289,073	6.16	11.63
	Reference	100	100	4,802,733	66.28	4,802,733	0	0	-
	Causalcall	NA	NA	130,962	42.95	13,522	NA	NA	NA
	Bonito_CRF-fast	0.002	92.36	119,570,537	40.34	368,848	2,860	0	18.87
	Bonito_CTC	0.430	95.06	134,732,516	40.86	371,590	384,243	0.29	18.58
Human HG002	SACall	NA	NA	63,025,520	39.87	320,873	NA	NA	NA
	Dorado-fast	0.001	93.15	121,146,376	39.8	361,677	926	0	17.46
	RUBICALL	0.125	94.50	140,928,248	40.99	393,950	100,256	0.1	17.81
	Reference	100	100	2,947,743,500	40.79	2,947,743,500	0	0	-

Third, based on these two new metrics, we added a new observation in Section 2.5.1 De Novo Assembly that RUBICALL consistently provides lower errors or structural variants.

Fifth, the low Total Indels and Indel Ratio (%) for RUBICALL in an assembled sequence signify a
sequence that closely resembles the expected reference with minimal insertions and deletions (indels).
This indicates a well-structured and high-quality assembly. Such assemblies offer a clear and accurate
representation of the original sequence, facilitating downstream analyses, gene prediction, functional
annotation, and comparative genomics.

*[IR3.11] What species are the training data from?*

To enhance reproducibility, we use an open-source ONT training dataset of ~12.2 GiB, which can
be downloaded through the official ONT Bonito repository [62] using the command ‘bonito download
–training’. The dataset comprises 1,221,470 reads, all sequenced from complete genomes. Although ONT
has made the dataset publicly available, the specific DNA species contained within it has yet to be disclosed.
A previous work [R3] calculated an approximate list of 496 unique taxonomic IDs using the Kraken2
taxonomic classification system in this ONT training dataset. To have a fair comparison, we use this ONT
dataset to train all the basecallers.

[R3] Larsen ACM, Knudsen CA, Hansen MN. Palamut - An Expansion of the Bonito basecaller using lan-
guage models. Thesis. 2020. Available at: [https://projekter.aau.dk/projekter/files/334904330/](https://projekter.aau.dk/projekter/files/334904330/MI104F20_Speciale___Paper__21_.pdf)
[MI104F20_Speciale___Paper__21_.pdf](https://projekter.aau.dk/projekter/files/334904330/MI104F20_Speciale___Paper__21_.pdf)

We updated details on training data in Section 5 **Methods** as follows:

The dataset comprises 1,221,470 reads, all sequenced from complete genomes. This ONT training
dataset has an approximate list of 496 unique taxonomic IDs using the Kraken2 [95] taxonomic
classification system [96].

*[IR3.12] I did not see where the explanation and application for hardware coupling and optimization*
*was. It is talked about in the paper, but I was unable to see the implementation or details. Could the*
*authors please clarify?*

Thank you for bringing up the need for a more detailed explanation and application of hardware coupling
and optimization in our framework.

In our QABAS method, hardware coupling and optimization are fundamental to achieving efficient
basecalling architecture tailored to specific hardware platforms. We employ a combination of hardware-
aware optimization techniques and customization strategies to ensure optimal performance. Here’s a more
detailed explanation of our approach:

**Hardware-aware Optimization Techniques:** We incorporate optimization techniques that consider
the unique characteristics and constraints of the target hardware. These techniques include memory usage,
computation capabilities, and bit-width optimization at each neural network layer. In doing so, we tailor
the neural network architecture and computation to align with the hardware’s capabilities.

**Customization for Target Hardware:** We provide users with the flexibility to customize the framework
for their target hardware (using the `applied_hardware` flag in RUBICON [127]) by adjusting hardware-
specific parameters. Different hardware provides different latencies for the same layers chosen from
the QABAS search space. The user can use the `reference_latency` flag in QABAS to guide the search of
basecalling architecture to find an architecture that meets certain latency constraints. This coupling of

target hardware latency ensures the basecaller architecture is finely tuned to operate optimally on the intended hardware.

We addressed this comment by updating Section **S1 Quantization-Aware Basecaller Architecture Search (QABAS)** as follows. First, we add the following to the benefit of using bit-width optimization to the QABAS search space.

In doing so, we tailor the neural network architecture and computation to align with the hardware’s capabilities.

Second, we provide details on the flexibility of RUBICON in Section **S1 Quantization-Aware Basecaller Architecture Search (QABAS)** as follows:

As different hardware provides different latencies for the same layers chosen from the QABAS search space, the user can customize the RUBICON framework for their target hardware by adjusting hardware-specific parameters (i.e., using the `applied_hardware` flag in RUBICON [127]). We provide an additional `reference_latency` flag in QABAS to guide the search of basecalling architecture to find an architecture that meets certain latency constraints. This coupling of target hardware latency ensures the basecaller architecture is finely tuned to operate optimally on the intended hardware.

[IR3.13] I appreciate the code being shared but will say that figshare is an odd choice. Is there a plan to also share via GitHub?

Thank you for your feedback regarding the sharing of code associated with our work. We have now successfully completed the AMD/Xilinx open-source process and have fully open-sourced our code at <https://github.com/Xilinx/neuralArchitectureReshaping/>.

[IR3.14] As a suggestion, I think a Jupyter notebook might be a nice addition to the codebase.

Thank you for this suggestion. We have added several notebooks and documentation on our open-source git repo (<https://github.com/Xilinx/neuralArchitectureReshaping/tree/main/notebook>). We plan to add more notebook examples on a rolling basis.

Second round of review

Reviewer 2

The authors have addressed all the questions of concern and implemented the necessary revisions.